# Evaluating Self-Supervised Learning for Molecular Graph Embeddings

**Hanchen Wang**[1⋆] **Jean Kaddour**[2⋆] **Shengchao Liu**[3,4⋆] **Jian Tang**[3,5,6] **Joan Lasenby**[1] **Qi Liu**[7]

[1]Cambridge    [2]UCL    [3]MILA    [4]UdeM    [5]HEC    [6]CIFAR    [7]HKU    ⋆ Equal Contribution

## Abstract

Graph Self-Supervised Learning (GSSL) provides a robust pathway for acquiring embeddings without expert labelling, a capability that carries profound implications for molecular graphs due to the staggering number of potential molecules and the high cost of obtaining labels. However, GSSL methods are designed not for optimisation within a specific domain but rather for transferability across a variety of downstream tasks. This broad applicability complicates their evaluation. Addressing this challenge, we present "Molecular Graph Representation Evaluation" (MOLGRAPHEVAL), generating detailed profiles of molecular graph embeddings with interpretable and diversified attributes. MOLGRAPHEVAL offers a suite of probing tasks grouped into three categories: (i) generic graph, (ii) molecular substructure, and (iii) embedding space properties. By leveraging MOLGRAPHEVAL to benchmark existing GSSL methods against both current downstream datasets and our suite of tasks, we uncover significant inconsistencies between inferences drawn solely from existing datasets and those derived from more nuanced probing. These findings suggest that current evaluation methodologies fail to capture the entirety of the landscape.

## 1  Introduction

Learning neural embeddings of molecular graphs has become of paramount importance in computer-aided drug discovery [1, 2]. For instance, a molecular property prediction (MPP) model can expedite and economise the design process by reducing the need for synthesising and measuring molecules. Thereby, such models can be immensely useful in the hit-to-lead and early lead optimisation phase of a drug discovery project [3]. However, obtaining labels of molecule properties is expensive and time-consuming, especially since the size of potential pharmacologically active molecules is estimated to be in the order of $10^{60}$ [4, 5].

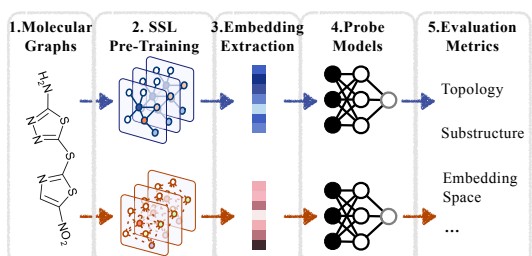

Figure 1: Overview of MOLGRAPHEVAL. Given molecular graphs, we train GNNs to predict SSL proxy objectives. We then extract embeddings of (possibly unseen) graphs using pre-trained models, which form the inputs for probe models, trained and evaluated on the designed metrics.

Graph Self-Supervised Learning (GSSL) paves the way for learning molecular graph embeddings without human annotations that are transferable to various downstream datasets. Unfortunately, the evaluation of such general-purpose embeddings is fundamentally complex. Different proxy objectives will place different demands on them, and no single downstream dataset can be definitive. Moreover, many of the previously proposed GSSL works are disconnected in terms of the tasks they target and the datasets they use for evaluation, making direct comparison difficult.

37th Conference on Neural Information Processing Systems (NeurIPS 2023) Track on Datasets and Benchmarks.

**Contributions.** Our goal is to unbiasedly evaluate molecular graph embeddings obtained by GSSL methods on existing downstream tasks and a new suite of probe tasks (Fig. 1). We summarised some key findings based on a total of **90,918 probe models** and **1,875 pre-trained GNNs**.

1) On MPP tasks, we observe every GSSL method can introduce substantial performance gains. Yet, there is a significant difference in the rank depending on whether we fine-tune the pre-trained network on the downstream dataset or not. Also, the pre-training configurations for obtaining the optimal embeddings or initialisation for fine-tuning are different; see - **Finding 2**.

2) Several discrepancies between MPP tasks and MOLGRAPHEVAL demonstrate how the latter complements GSSL evaluation with novel insights fostering future work:
   - While embeddings from a randomly initialised GNN perform poorly on MPP tasks, they sometimes outperform on topological properties (which can be useful in some molecular tasks), indicating that GSSL methods are not universally better, see - **Finding 4**.
   - While contrastive GSSL methods perform better than most other methods on MPP tasks, it might attribute to their superiority in identifying the crucial substructures, see - **Finding 8**.
   - In contrast to previous work [6], we find that feature distribution uniformity is not always a strong indicator for MPP performance. For instance, maximising the mutual information between multi-scale graph representations (INFOGRAPH) results in the most uniform distributions, yet, it ranks $7^{\text{th}}$ among the 9 GSSLs on the downstream MPP tasks, see - **Finding 9**.

## 2    Related work

**Graph SSL (GSSL)** can be divided into contrastive and generative methods [7–9]. Contrastive GSSL [10–12] construct multiple views of the same graph via augmentations and then learn embeddings by aligning positive samples against negative ones. Generative GSSL [11, 13–15] yields embeddings by reconstructing input graphs. Zhu et al. [16] conduct an empirical analysis of contrastive GSSL methods and their components. In contrast, we investigate both generative and contrastive GSSL methods and propose a novel suite of tasks to probe the learned embeddings' attributes.

**Probe models and benchmarks on graphs.** Probe models, trained exclusively on embedding vectors from pre-trained models, serve as an effective tool for evaluating the quality of learned embeddings [17]. Their effectiveness has been demonstrated across various domains such as language [18–23], vision [24–28], relational tables [29], and science [30–32]. While there exist benchmarks for graph learning [33–37], applying probe models to GSSL remains an unexplored frontier.

## 3    Preliminaries

**Graph.** A graph $\mathcal{G} = (\mathcal{V}, \mathcal{E})$ consists of a set of nodes $\mathcal{V}$ and edges $\mathcal{E}$. In molecular graphs, nodes are atoms, and edges are bonds. We use $\boldsymbol{x}_u$ and $\boldsymbol{x}_{uv}$ to denote the feature of node $u$ and the bond feature between nodes $[u, v]$, respectively. For notation simplicity, we use an adjacency matrix $\mathbf{A} \in \mathbb{R}^{|\mathcal{V}| \times |\mathcal{V}|}$ to represent the graph, where $\mathbf{A}[u, v] \neq 0$ if the nodes $(u, v)$ are connected.

**GNN.** Graph neural networks (GNNs) give rise to learning molecular graph embeddings [14, 38–41]. A prototypical GNN relies on messaging passing [39], which updates atom-level embeddings based on their neighbourhoods. Given an input atom $\boldsymbol{h}_u^0 = \boldsymbol{x}_u$, we compute its embedding by:

$$\boldsymbol{m}_u^{t+1} = \sum_{v : \mathbf{A}[u,v] \neq 0} M_t(\boldsymbol{h}_u^t, \boldsymbol{h}_v^t, \boldsymbol{x}_{uv}) \qquad \boldsymbol{h}_u^{t+1} = U_t(\boldsymbol{h}_u^t, \boldsymbol{m}_u^{t+1}) \qquad (1)$$

where $M_t$ and $U_t$ are the "message" functions and "vertex update" functions, respectively. Repeating message passing for $T$ steps, the embedding of each atom contains their $T$-hop neighbourhood information. A readout function $R$ is then used to pool node-level embeddings for graph-level representations: $\hat{y} = R(\{\boldsymbol{h}_u^\top \mid u \in \mathcal{V}\})$. Following previous GSSL methods on molecular graphs, we adopt the Graph Isomorphism Network (GIN) [42] as the backbone model and incorporate edge features during message passing following [11].

**Pre-Training.** We inspect nine GSSL methods (1,875 configurations in total): EDGEPRED [13], INFOGRAPH [10], GPT-GNN [15], ATTRMASK [11], CONTEXTPRED [11], GROVER [43], GRAPHCL [12], JOAO [44], and GRAPHMVP [45]. We use all qualified molecules (around 0.33 million, *i.e.*, leave out the molecules that appeared in downstream datasets) from the GEOM dataset [46] to pre-train the GIN backbone. As many of these pre-training methods are not primarily designed for molecular graphs, we perform the grid search over the hyperparameter space and save

Table 1: **Evaluating GSSL methods on molecular property prediction tasks.** For each downstream dataset, we report the mean and standard deviation of the ROC-AUC scores over three random scaffold splits. The best and second best scores are marked **bold** and **bold**, respectively. The performance scores are based on the fixed pre-trained embeddings with linear probe models, we also report the average ROC-AUC scores with fine-tuned pre-trained GNN on MPP tasks ("Avg (FT)"). For each pre-training method, we report the highest scores in the table and their corresponding hyperparameter configurations in Tables 7 and 9 in Appendix B.

| | BBBP | Tox21 | ToxCast | Sider | ClinTox | MUV | HIV | Bace | Avg | Avg (FT) |
|---|---|---|---|---|---|---|---|---|---|---|
| # Molecules | 2,039 | 7,831 | 8,575 | 1,427 | 1,478 | 93,087 | 41,127 | 1,513 | – | – |
| # Tasks | 1 | 12 | 617 | 27 | 2 | 17 | 1 | 1 | – | – |
| RANDOM | $50.7_{\pm2.5}$ | $64.9_{\pm0.5}$ | $53.2_{\pm0.3}$ | $53.2_{\pm1.1}$ | $63.1_{\pm2.3}$ | $62.1_{\pm1.3}$ | $66.1_{\pm0.7}$ | $63.4_{\pm1.8}$ | 59.60 | 66.16 |
| EDGEPRED | $54.2_{\pm1.0}$ | $66.2_{\pm0.2}$ | $54.4_{\pm0.1}$ | $56.1_{\pm0.1}$ | $\mathbf{65.4}_{\pm5.0}$ | $59.5_{\pm0.9}$ | $\mathbf{73.6}_{\pm0.4}$ | $71.4_{\pm1.2}$ | 62.59 | 68.16 |
| ATTRMASK | $62.7_{\pm2.7}$ | $65.7_{\pm0.8}$ | $\mathbf{56.1}_{\pm0.2}$ | $58.3_{\pm1.5}$ | $61.9_{\pm6.4}$ | $60.9_{\pm1.8}$ | $65.5_{\pm1.4}$ | $64.8_{\pm2.6}$ | 61.99 | 69.20 |
| GPT-GNN | $62.0_{\pm0.9}$ | $64.9_{\pm0.7}$ | $55.4_{\pm0.2}$ | $55.3_{\pm0.8}$ | $55.0_{\pm5.1}$ | $61.2_{\pm1.5}$ | $71.2_{\pm1.5}$ | $61.0_{\pm1.2}$ | 60.74 | 67.58 |
| INFOGRAPH | $65.9_{\pm0.6}$ | $65.8_{\pm0.7}$ | $54.6_{\pm0.1}$ | $57.2_{\pm1.0}$ | $61.4_{\pm4.8}$ | $\mathbf{63.9}_{\pm1.9}$ | $71.4_{\pm0.6}$ | $67.4_{\pm4.9}$ | 63.44 | 68.92 |
| CONT.PRED | $55.5_{\pm2.0}$ | $67.9_{\pm0.7}$ | $54.0_{\pm0.3}$ | $57.1_{\pm0.5}$ | $\underline{\mathbf{67.4}}_{\pm4.3}$ | $60.5_{\pm0.9}$ | $66.2_{\pm1.5}$ | $54.4_{\pm3.2}$ | 60.36 | 69.40 |
| GROVER | $\mathbf{67.0}_{\pm0.3}$ | $63.9_{\pm0.3}$ | $53.6_{\pm0.4}$ | $\underline{\mathbf{59.9}}_{\pm1.7}$ | $65.0_{\pm6.4}$ | $62.7_{\pm1.4}$ | $67.8_{\pm1.0}$ | $69.0_{\pm4.7}$ | 63.62 | 69.97 |
| GRAPHCL | $64.7_{\pm1.7}$ | $\underline{\mathbf{69.1}}_{\pm0.5}$ | $\underline{\mathbf{56.2}}_{\pm0.2}$ | $59.5_{\pm0.9}$ | $60.8_{\pm3.0}$ | $60.6_{\pm1.8}$ | $72.5_{\pm1.4}$ | $\underline{\mathbf{77.0}}_{\pm1.7}$ | **65.04** | **70.33** |
| JOAO | $66.1_{\pm0.8}$ | $\mathbf{68.1}_{\pm0.2}$ | $55.1_{\pm0.4}$ | $58.3_{\pm0.3}$ | $65.3_{\pm6.1}$ | $62.4_{\pm1.2}$ | $\underline{\mathbf{73.8}}_{\pm1.2}$ | $71.1_{\pm0.8}$ | **65.05** | 69.75 |
| GRAPHMVP | $\underline{\mathbf{69.2}}_{\pm1.8}$ | $63.8_{\pm0.3}$ | $55.5_{\pm0.3}$ | $58.6_{\pm0.4}$ | $58.7_{\pm1.9}$ | $\mathbf{63.8}_{\pm1.3}$ | $68.6_{\pm1.0}$ | $\mathbf{73.3}_{\pm4.7}$ | 63.92 | **70.06** |

the optimal settings. For these nine GSSL methods, we have pre-trained 1,875 GNNs with different configurations, as elaborated in Appendix B. We extract embeddings using the pre-trained weights, select the optimal hyperparameter sets based on their downstream MPP performance and use these optimal embeddings for further probing tasks.

**Probe.** We use probe models [18] to study whether self-supervised learned embeddings encode helpful structural information about graphs. Concretely, we extract embeddings from a pre-trained GNN and train a linear model to predict the probe tasks with node and graph embeddings as inputs. As the first work that designs probe methods on graph embeddings, we follow previous works on computer vision and natural language processing. We mainly compute and compare the quality of pre-trained embeddings using linear probe models. We have also experimented MLPs with one hidden layer as the probe models, as this architecture is utilised in some previous works. We observe similar findings with both probe architectures and reported the results of MLP probes in Appendix B.4. We use scaffold split to partition data into 80%/10%/10% for the training/validation/testing set. The training procedure runs for 100 epochs with a fixed learning rate of 0.001. We report the test results based on the best validation scores. To account for statistical significance, we average all experimental results over three independent runs. We find that different data splits are the primary cause for performance variations (∼2%), instead of initialising probe models with different random seeds (<0.01%).

## 4 Benchmarking GSSL on MPP

We first conduct a rigorous empirical investigation of the GSSL methods' effectiveness in predicting the biochemical properties of molecules. Following previous work [11, 12], we consider eight molecular datasets consisting of 678 binary property prediction tasks [47, 48]. Unless explicitly stated otherwise, we extract the node/graph embeddings from the last GNN layer. We devise two settings: (i) fixed embeddings, where we train the probe models with fixed embeddings extracted from pre-trained GNNs; (ii) fine-tuned embeddings ("FT"), where we update weights of both the pre-trained GNNs and the probe models. Setting (i) follows the procedures in previous probing literature, while (ii) is widely utilised as the "pre-training, then fine-tuning" paradigm. We use Adam optimiser with no weight decay, set the batch size as 256, and apply identical pre-processing procedures for all experiments.

**Findings.** Table 1 notes the results, and we summarise the following findings, some of which contrast with those drawn from the concurrent study [49].

1) **All GSSL methods perform better than RANDOM.** By carefully exploring the pre-training hyperparameters, all GSSLs substantially improve the MPP tasks for both fixed and fine-tuned embeddings. Contrastive-based GSSL methods (*i.e.*,, GRAPHCL, JOAO and GRAPHMVP) achieve the overall best performance. As [49] declares that molecular graph pretraining is ineffective; however, we find that their conclusions are based on a few selected finetuning datasets and fixed

pre-training hyperparameters. We further observe that such improvements will reduce when the number of molecules in downstream datasets increases. Specifically, for MUV [50], a dataset designed for validating virtual screening (used in drug discovery to find how likely molecules that bind to a drug target), the average performance gain brought by pre-training is -0.3%; while for BBBP, it is 12.3%. The number of molecules in BBBP is only 2% of the MUV's.

2) **Rankings differ between probing and fine-tuning.** The rank correlation between the fixed and fine-tuned embeddings is 0.77 (p-value=9e-4), indicating that we cannot utilise the rank of fixed embeddings as a definite indicator for fine-tuning performance, though they are positively correlated. Part of this observation has been spotted in a study on masked visual transformers [25]. In the context of molecular property prediction, embeddings pre-trained with JOAO achieve the best score with fixed scenarios but perform the fourth after end-to-end fine-tuning. The reason is unclear and should be investigated by future work.

3) **The optimal sets of pre-train hyperparameters for fixed and fine-tuned embeddings vary.** We observe that the optimal pre-training hyperparameters on fixed and fine-tuned embeddings differ. Only two out of nine GSSLs (INFOGRAPH and EDGEPRED) share the same set of optimal parameters, as detailed in Tables 7 and 9 in Appendix B. This suggests that probing the fixed embeddings might not truly reflect pre-trained models' performance on downstream MPP tasks, as it ignores the consequent improvements induced by fine-tuning. In Fig. 2, we visualised the hyperparameter space of the ATTRMASK pre-trainer, the local minima in the hyperparameter space distribute differently. As shown in Fig. 2 also Table 9, the best pre-training configuration for probing is "mask rate=0.85 and learning rate=1e-4", while in terms of fine-tuning scores, the optimal setting is "mask rate=0.50 and learning rate=5e-4". Also, it can be inferred that the optimal pre-training hyperparameters for different pre-training datasets vary; therefore, using reported hyperparameters without carefulness and concluding "graph pretraining is ineffective in molecular domain" is not convincing [49].

## 5 Molecular graph representation evaluation

The goal of GSSL for molecular graphs is to obtain embeddings that capture generic information about the molecule and its properties. However, there is no free lunch [51]: different training objectives optimise for different properties, and evaluating the extracted embeddings on only a handful of downstream datasets does not provide the whole picture (as we confirm empirically in Sec. 6). Also, from Sec. 4, we know the probing and fine-tuning performance are positively correlated, yet their optimal pre-training configurations are largely diverging. In the Appendix, we also provide results on the worst pre-training configurations, some of which

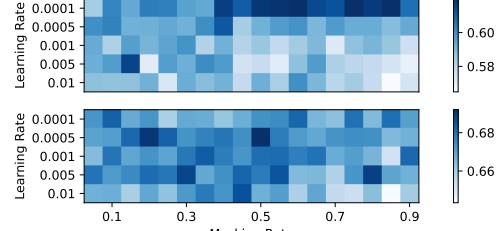

Figure 2: **Hyperparameter space of ATTR-MASK** (mask rate × learning rate), coloured by MPP test scores. Above: fixed; below: fine-tuned. Note the colour bars are different.

cause negative transfer due to initialising the encoders into local bad minima. Investigations are required to understand what kind of property makes the pre-trained encoders differ.

To this end, we propose MOLGRAPHEVAL, which encompasses a variety of carefully-selected probe tasks, categorised into three classes: (i) **generic graph properties**, (ii) **molecular substructure properties** and (iii) **embedding space properties**. In the upcoming subsections, we explain the tasks in more detail and why they are essential for molecular graph embeddings.

### 5.1 Generic graph properties

Topological property statistics are often used as features in machine learning pipelines on graphs that do not rely on neural networks [52]. Based on their scale, they can be divided into {node-, pair-, and graph-} level statistics. For molecular graphs, topological metrics have been widely used as molecular descriptors in cheminformatics for decades [53–56], metrics at different scales will facilitate different tasks.

**Node-level statistics** accompany each node with a local topological measure, which could be used as features in node classification [52]. Concretely, node-level information such as degree [57] can reflect the reaction centres [58]; thus, it can aid in discovering chemical reactions [59].

- **Node degree** ($d_u$) counts the number of edges incident to node $u$: $d_u = \sum_{v \in V} \mathbf{A}[u, v]$.
- **Centrality** ($e_u$) represents a node's importance. The eigenvector centrality is determined by a relation proportional to the average centrality of its neighbours: $e_u = \left( \sum_{v \in V} \mathbf{A}[u, v] e_v \right) / \lambda$, $\forall u \in \mathcal{V}$.
- **Clustering coefficient** ($c_u$) measures how tightly clustered a node's neighbourhood is: $c_u = (|\ (v_1, v_2) \in \mathcal{E} : v_1, v_2 \in \mathcal{N}(u)|) / d_u^2$, *i.e.*, the fraction of closed triangles in neighbourhood [60].

**Graph-level statistics** summarise global topology information and are helpful for graph classification tasks. For molecules, graph-level statistics can be used, *e.g.*, to classify a molecule's solubility [61]. We briefly describe their intuitions; formal definitions can be found, *e.g.*, in [52].

- **Diameter** is the maximum distance between the pair of vertices (*i.e.*, longest path in a molecule).
- **Cycle basis** is a set of simple cycles that forms a basis of the graph cycle space. It is a minimal set that allows every even-degree subgraph to be expressed as a symmetric difference of basis cycles.
- **Connectivity** is the minimum number of elements (nodes or edges) that need to be removed to separate the remaining nodes into two or more isolated subgraphs.
- **Assortativity** measures the similarity of connections in the graph with respect to the node degree. It can be seen as the Pearson correlation coefficient of degrees between pairs of linked nodes.

**Pair-level statistics** quantify the relationships between nodes (atoms), which is vital in molecular modelling. For example, molecular docking techniques aim to predict the best matching binding mode of a ligand to a macro-molecular partner [62]. For predicting such binding compatibility, connectivity and distance awareness (how close a pair of atoms can be) are important. In our implementation, we randomly select a fixed number (*i.e.*,, 10) of atom pairs from each molecular graph. These pairs are then categorised based on their originating molecule, ensuring all pairs from a single molecule are designated to a singular split: either train, validation, or test.

- **Link prediction** tests whether two nodes are connected or not, given their embeddings and inner products. Based on the principle of *homophily*, it is expected that embeddings of connected nodes are more similar compared to disconnected pairs:

$$\mathbf{S}_{\text{Link}}[u, v, \mathbf{x}_u^T \mathbf{x}_v] = \mathbb{1}_{\mathcal{N}(u)}(v) \tag{2}$$

- **Jaccard coefficient** seeks to quantify the overlap between neighbourhoods while minimising the biases induced by node degrees [63]:

$$\mathbf{S}_{\text{Jaccard}}[u, v] = |\mathcal{N}(u) \cap \mathcal{N}(v)| / |\mathcal{N}(u) \cup \mathcal{N}(v)| \tag{3}$$

- **Katz index** is a global overlap statistic defined by the number of paths between a pair of nodes:

$$\mathbf{S}_{\text{Katz}}[u, v] = \sum_{i=1}^{\infty} \beta^i \mathbf{A}^i[u, v] \tag{4}$$

where $\beta \in \mathbb{R}^+$ determines the weight between short and long paths. $\beta < 1$ reduces the weight of long paths, in implementations we set $\beta = 1$ to give all paths equal importance.

## 5.2 Molecular substructure properties

Table 2: **ROC-AUC scores of classifiers predicting molecular properties**.

| Linear Regression | Random Forest | XGBoost | Random (FIX/FT) | JOAO (FIX) | GraphCL (FT) |
|---|---|---|---|---|---|
| 59.91 | 61.95 | 62.31 | 59.60 / 66.16 | 65.05 | 70.33 |

Molecular substructures often serve as reliable indicators of biochemical properties [64–67]. For instance, molecules with benzene rings typically share consistent physical properties, such as solubility, as well as chemical characteristics like aromaticity [68].

**Substructures.** We investigate 24 substructures from three groups: **rings** (Benzene, Beta lactams, ..., Thiophene); **functional groups** (Amides, Amidine, ..., Urea); and **redox active sites** (Allylic). We provide chemical knowledge on how they relate with molecular properties in Appendix D.

**How predictive are substructures?** To demonstrate that substructures are quite predictive of molecular properties, we utilise counts of substructures within a molecular graph as the input for

classic ML methods (linear regression, random forest, and XGBoost) to predict the molecular properties on eight downstream datasets. Table 2 shows the results. For ease of comparison, we add the performance of RANDOM, GRAPHCL (FIX), and JOAO (FT) from Table 1. Notably, even basic models trained exclusively on substructures yield performance akin to the randomly trained GNN baseline. This underscores the profound correlation between substructures and MPP task efficacy.

## 5.3 Embedding space properties

Beyond MPP metrics, we assess domain-agnostic properties of the embedding space produced by pre-trained graph encoders. These properties correlate positively with downstream generalisation [6]. Hence, they can serve as proxies for embedding degeneration, especially in label-scarce scenarios. We explore three such properties in MOLGRAPHEVAL:

- **Alignment** quantifies how similar produced embeddings are for similar samples [6]. Ideally, two samples with the same (or very similar) semantics should be mapped to nearby features, thus mostly invariant to unneeded noise factors. To examine alignment, we construct positive and negative molecule pairs. Positive pairs in a dataset are those that share identical molecular properties, whereas negative pairs are those that differ in their properties. A better alignment represents a better nearest neighbourhood formulation, which has been especially useful for tasks such as compound potency prediction [70].
- **Uniformity** measures how uniformly the embeddings are distributed on the unit hypersphere [6]. A more uniformly distributed embedding space is expected to be with better generalisation under some mild assumptions.
- **Dimensional collapse** refers to the problem of embedding vectors spanning a lower-dimensional subspace instead of the entire available embedding space [69, 71]. Following Jing et al. [69], one simple way to test the occurrence of dimension collapse is to inspect the number of non-zero singular values of a matrix stacking the embedding vectors $\mathbf{Z} = \{\mathbf{z}_i\}_{i=1}^N$.

Figure 3: Embedding space property.

(a) **Alignment** [6]

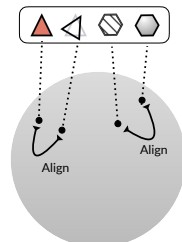

(b) **Uniformity** [6]

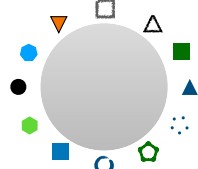

(c) **Dimensional collapse** [69]

## 6 Results

### 6.1 Generic graph properties

For node- and graph-level topological properties, we benchmark GSSL methods on all the molecular graphs from each dataset; for pair-level metrics, we bootstrap 10k node pairs from each dataset for evaluation. The reported test scores are averaged over three runs. We plot the distribution of these metrics in Appendix E.

**Findings.** Table 3 shows the results, and we summarise these findings.

4) **RANDOM outperforms almost every GSSL method on node- and pair-level metrics**, therefore incorporating the randomised features would bring substantial advantages for tasks that are based on local geometry [72]. We observe that among six node- and pair-level graph metrics, randomised embeddings perform the best in four. We further verify this via T-SNE embeddings in Fig. 4, where randomised embeddings can form more interpretable clusters w.r.t. "Atom Nodes". In Appendix C, we provide a numerical analysis to show that one of the reasons might be the

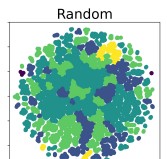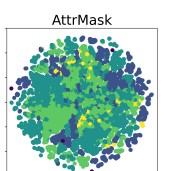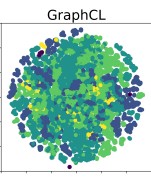

Figure 4: **RANDOM generates more interpretable latent space**: T-SNE visualisation of node embeddings produced by RANDOM, ATTRMASK, and GRAPHCL on BBBP. Each dot represents an atom from a molecular graph, coloured by node degrees. We notice that RANDOM embeddings form more coherent clusters, consistent with their performance on node-level tasks. While this behaviour is not observed in other atom-level metrics such as node centrality (Fig. 8).

choice of the GNN layer's initialisation and summation message-passing function. We further empirically experiment with different initialisation strategies, finding that the discriminative power of randomised embeddings on these local metrics disappears.

5) **RANDOM falls short of predicting graph-level metrics**, which is in alignment with the fact that all GSSL methods outperform RANDOM on the graph-level MPP tasks. However, ranking top on recovering graph topological metrics does not guarantee the embeddings have the best generalisation on downstream MPP tasks.

6) **Performance in topological metrics aligns with pre-training objectives.** EDGEPRED and ATTRMASK set the pre-training objectives to predict the adjacency matrix and masked nodes and edges, respectively; the embeddings extracted from these methods compose more information on the node- and pair-level topological metrics. Also, incorporating 3D geometry in the pre-training (*i.e.*, GRAPHMVP) helps retain pair-level topological properties. It aligns with the fact that geometry helps improve target discovery [73–75], as atom pair interactions play a major role.

Table 3: **Benchmarking topological properties.** We report the mean square error or the cross entropy on eight datasets (*i.e.*, smaller is better). For each topological metric, the best and second-best scores are marked **bold** and **bold**, respectively. We omit such annotations for the "Clustering coefficient (*i.e.*, Cluster)" metric as all test errors are similar. We also report the average ranks of each GSSL method on these metrics ("R"), grouped by the three abstract levels of topology. We use the pre-training configurations that achieve the best MPP test performance with fixed embeddings, as reported in Tables 1 and 9.

| | Node | | | | Pair | | | | Graph | | | | |
|---|---|---|---|---|---|---|---|---|---|---|---|---|---|
| | Degree | Cent. | Cluster | R | Link | Jaccord | Katz | R | Diameter | Conn. | Cycle | Assort. | R |
| RANDOM | **0.001** | **0.008** | 0.003 | **1.5** | 0.078 | **0.012** | **0.017** | 2.8 | 177.924 | 0.087 | 2.933 | 0.029 | 8.6 |
| EDGEPRED | 0.031 | 0.009 | 0.003 | 3 | **0.067** | **0.014** | **0.016** | **2.2** | 159.825 | 0.073 | 2.596 | 0.026 | 6.5 |
| ATTRMASK | **0.009** | 0.009 | 0.003 | 2.7 | 0.082 | 0.015 | 0.020 | 4.7 | 110.793 | **0.062** | 2.207 | **0.019** | **2** |
| GPT-GNN | 0.123 | 0.009 | 0.003 | 4.7 | **0.014** | 0.021 | 0.029 | 5.3 | 111.688 | 0.074 | 2.854 | 0.026 | 6.5 |
| INFOGRAPH | 0.054 | 0.009 | 0.004 | 5 | 0.088 | 0.019 | 0.021 | 6 | **84.339** | 0.066 | **2.100** | 0.029 | 6 |
| CONT.PRED | 0.164 | 0.010 | 0.004 | 8 | **0.014** | 0.021 | 0.045 | 5.7 | 138.304 | 0.067 | 2.150 | 0.027 | 5.3 |
| GROVER | 0.120 | 0.012 | 0.004 | 8.2 | 0.114 | 0.047 | 0.059 | 10 | **78.352** | 0.064 | **2.058** | **0.021** | **3.3** |
| GRAPHCL | 0.060 | 0.010 | 0.004 | 6.7 | 0.084 | 0.028 | 0.026 | 7.3 | 90.336 | 0.066 | 2.287 | 0.026 | 6.1 |
| JOAO | 0.067 | 0.010 | 0.004 | 7 | 0.089 | 0.041 | 0.025 | 8 | 95.335 | **0.063** | 2.352 | 0.024 | 5.3 |
| GRAPHMVP | 0.199 | 0.010 | 0.004 | 8.3 | 0.077 | **0.014** | **0.017** | 3 | 109.198 | 0.065 | 2.372 | 0.030 | 5.5 |

## 6.2 Substructure properties

We use Cramér's V statistics to identify the five substructures mostly associated with downstream biochemical properties, as reported in Table 5, detailed in Table 14. We probe the pre-trained embeddings to predict these substructures. We provide the complete results and plot the distributions of all substructures in Appendix D.

**Findings.** Table 4 shows the results, where we **bold** the best and underline the worst scores of each substructure. We report the Spearman rank correlation and p-values between the performance of recognising substructures and predicting molecular properties. We highlight the following findings.

7) **Substructure detection performance correlates well with MPP performance.** Pre-trained embeddings notably surpass the RANDOM in both substructure detection and Multiple Property Prediction (MPP)

Table 4: **Benchmarking substructure properties.** We report the mean square errors on the test split averaged over the eight downstream MPP datasets.

| | allylic | amide | benzene | ether | halogen |
|---|---|---|---|---|---|
| RANDOM | 0.959 | 16.917 | 1.100 | 2.024 | 1.127 |
| EDGEPRED | 0.780 | 14.173 | 0.797 | 1.608 | 0.939 |
| ATTRMASK | 0.926 | 14.703 | 0.976 | 1.742 | 0.501 |
| GPT-GNN | 0.872 | 15.629 | 0.783 | 1.912 | 0.341 |
| INFOGRAPH | 0.740 | 6.747 | 0.583 | 1.128 | 0.706 |
| CONT.PRED | 1.040 | 16.636 | 0.980 | 1.787 | 1.075 |
| GROVER | 0.715 | **6.576** | 0.558 | **0.957** | **0.298** |
| GRAPHCL | **0.652** | 7.598 | **0.525** | 1.077 | 0.319 |
| JOAO | 0.654 | 7.926 | 0.531 | 1.071 | 0.310 |
| GRAPHMVP | 0.905 | 6.992 | 0.649 | 1.037 | 0.311 |
| Corr. | 0.830 | 0.770 | 0.915 | 0.879 | 0.806 |
| p-value | 3e-3 | 9e-3 | 2e-4 | 8e-4 | 5e-3 |

Table 5: **Cramér's V between molecular substructure counts and binary properties**, averaged over 678 property prediction tasks or eight downstream datasets.

| | allylic | benzene | amide | ether | halogen |
|---|---|---|---|---|---|
| Task | 0.1144 | 0.1630 | 0.0881 | 0.1034 | 0.1721 |
| Dataset | 0.1024 | 0.1227 | 0.1336 | 0.1083 | 0.1086 |

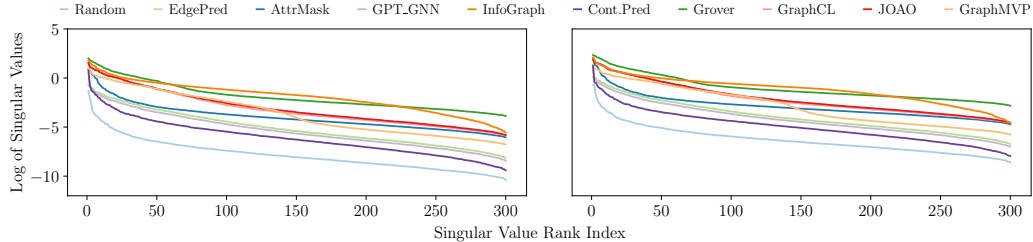

Figure 5: **Dimensional collapse analysis on BBBP**. Left: graph; Right: node.

tasks, with the sole exception of continuous prediction in the "allyli" category. The superior performance of GSSLs in predicting molecular properties could be attributed to their capacity for substructure awareness. This observation is further corroborated by the high positive rank correlation and substantial statistical significance, as indicated by all p-values being less than 1%. This evidence suggests that integrating substructure awareness into GSSL methods could potentially enhance the accuracy of molecular property predictions.

8) **Motif-based and Contrastive-based GSSL methods have better substructure awareness.** GROVER, GRAPHCL, JOAO, and JOAOV2 perform consistently better than most other GSSL methods on substructure detection also in the MPP tasks. Note that the optimal pre-training configurations for GROVER is "Motif"-based loss[1], as reported in Table 9.

### 6.3 Embedding space properties

**Findings.** Based on the above results, we summarise the following findings. For alignment, we randomly select 10k positive and negative pairs of molecular graphs from BBBP and Tox21 datasets, calculate the cosine similarity and plot the histogram in Fig. 6. We choose AT-

Table 6: **Benchmarking embedding space properties.** We report the rank correlations and p-values.

|  | Node Embed | Graph Embed | Uniformity |
|---|---|---|---|
| Correlation | 0.806 | 0.927 | 0.842 |
| p-value | 4e-3 | 6e-3 | 2e-3 |

TRMASK and GRAPHCL to represent generative and contrastive GSSL methods, respectively. Table 16 presents uniformity values as defined in [6]; Fig. 5 plots the magnitude of the singular values in the logarithm scale provided in Table 15.

9) **Compared with the RANDOM initialised GNN, GSSL methods give rise to better alignments, promote more uniform features and lift the spectrum.**
GSSL embeddings form distinguishable distributions for positive/negative pairs, while the RANDOM embeddings do not (a phenomenon often referred to as over-smoothing [76], see Fig. 6). All the GSSL methods have better uniformly distributed embeddings on all datasets (in Table 16). However, we found that a better alignment is not necessary to achieve better generalisation for the domain of the molecular graph ( Table 16). The singular values of stacked GSSL embeddings (both node and graph) are larger than RANDOM's by multiple magnitudes; also, we observe that the magnitude of the spectrum positively correlates with the downstream MPP performance (in Tables 6 and 15).

## 7 Conclusion

In this work, we challenged common practices in evaluating graph self-supervised learned embeddings of molecular graphs. First, we extensively searched the optimal hyperparameters and evaluated GSSL methods on common molecular property prediction tasks in an unbiased and controlled manner. Next, we presented MOLGRAPHEVAL, which is a diverse collection of probe tasks divided into three categories: (i) topological properties, (ii) substructure properties, and (iii) embedding space properties. Then we evaluated GSSL methods on MOLGRAPHEVAL and found surprising insights not revealed by the evaluation of MPP tasks alone.

---

[1]Here the concepts of "Motif" and "Substructure" are identical, under the context of molecular graphs.

The purpose of this work is to complement current evaluation practices with probe tasks and metrics that reveal novel insights, rather than arguing about which combinations of pre-training tasks yield the best downstream performance. Also, as our primary focus is the pre-trained GNN encoders, we leave the investigations of comparing probing and fine-tuning embeddings in the future. Our empirical findings suggest that there are many open questions on how to learn robust molecular graph embeddings without labels and a better understanding of these, along with a new methodology for solving some of the issues mentioned earlier (*e.g.*, dimensional collapse), are yet to come. Nevertheless, we are optimistic that the tasks proposed in this paper will benefit the GSSL research community to tackle these challenges and applied scientists in fields like drug discovery to yield additional insights that can help their problem.

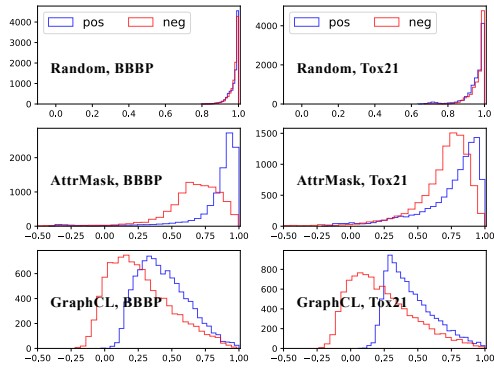

Figure 6: Alignment in the embedding space.

## Acknowledge

We thank Le Song, Anima Anandkumar, Matthew Welborn for valuable discussions.

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

# Appendix

## Table of Contents

# A  Overview

**Automated MOLGRAPHEVAL pipeline.** In the codebase of MOL-GRAPHEVAL, we provide setup scripts (in "env/") for both the docker and conda virtual environment. The end-to-end benchmarking pipeline consists of three modules (in Fig. 7):

- Pre-training the GNN models (other GNN model classes such as MPNN, GCN, GAT are implemented in the codebase);

- Extracting the node/graph/pair-level embeddings from the pre-trained or the randomly initialised GNN models;

- Probing the quality of embeddings with the proposed metrics.

We have meticulously packaged the components of MOL-GRAPHEVAL for ease of access and potential extension. The pre-trained methods are housed in "src/pretrainers", model libraries in "src/models", pre-training and downstream datasets in "src/datasets", and probing metrics in "src/validation". This modular design allows flexibility and extensibility to incorporate new model architectures and datasets. All configurations, specified in YAML files and processed by argument parsers in "src/config", are managed by scripts (with templates provided in "script"). Furthermore, we have implemented loggers to chronicle training/validation/testing curves during both pre-training and probing. For added convenience, templates for visualising embeddings and analysing datasets are also available.

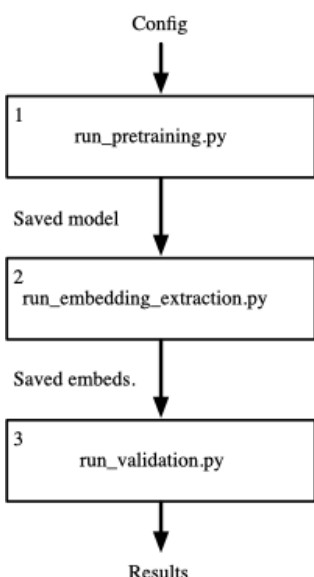

Figure 7: MOLGRAPHEVAL.

We open-sourced MOLGRAPHEVAL in https://github.com/hansen7/MolGraphEval.

# B  Pre-Training

We next describe the additional details of the pre-training used in the MOLGRAPHEVAL benchmark.

## B.1  GEOM dataset

We avoid using the updated version of GEOM ('New drug-like molecules' and 'MoleculeNet') to remove the overlap between pre-training and downstream datasets. Compared with other molecular datasets that contain 3D conformation structures, GEOM [46] has the following advantages:

- **Preciseness.** Compared with toolkits like RDKIT or MMFF [77] (used in studies such as ChemRL-GEM [78]), DFT-based calculations (used in GEOM) will provide more precise computation results on the 3D molecular conformation structures [79–83]. Such errors in the molecular geometries have been proven harmful to property predictions (Appendix B of [84])

- **Comprehensive.** GEOM provides a more comprehensive collection in comparison with other quantum chemistry-based datasets (*e.g.*, QM9 [85], Atom3D [86]) in terms of quantity and diversity. In comparison with ZINC15 and SAVI used in this concurrent study [49], GEOM provides accurate 3D conformation structures of molecules, which allows to compare more GSSL methods such as GraphMVP.

## B.2  GSSL methods

Self-Supervised Learning (SSL) generally bifurcates into contrastive and generative methodologies, each characterised by its unique supervised signal as highlighted by [87]. Contrastive SSL operates by contrasting representations at the inter-data level, while generative SSL emphasises intra-data level reconstruction. Both strategies have been the subject of comprehensive research.

**Contrastive GSSL** creates multi-layered views of each graph, each capturing different granularity levels, from nodes and subgraphs to the complete graph. It aligns representations for views originating from the same data and distinguishes those from unrelated datasets, targeting a unified

embedding space. The effectiveness of various approaches largely hinges on their view design. For example, INFOGRAPH contrasts node and graph views, while GRAPHCL and JOAO delve into graph-level transformations. Further avenues to improve Contrastive GSSL include:

- CLEAR [88] captures graph structure at both global and local scopes, enhancing semantic information granularity and consistency between multiple views;
- PGCL [89] addresses sampling bias by clustering graphs into groups represented by prototype vectors, focusing on the dataset's global semantics;
- iGCL [90] employs a Siamese architecture to generate positive samples sans data augmentation. The proposed ID loss eschews negative sampling while promoting feature-wise discriminability.

**Generative GSSL** zeroes in on reconstructing the graph structure, striving to derive representations that capture the core characteristics of the data. Noteworthy examples in this category include EDGEPRED and ATTRMASK, which predict adjacency matrices and mask tokens, respectively. Meanwhile, GPT-GNN employs an auto-regressive approach tailored for holistic graph reconstruction. In line with this methodology, masked graph autoencoders, as cited in [91–93], have garnered considerable attention.

GROVER-Motif leverages domain-specific knowledge to extract motifs from molecules, assigning SSL the role of predicting motif presence. Diverging from the paradigms of contrastive and generative GSSL, recent explorations like [94] categorise this approach as predictive GSSL. In this framework, the supervisory signals are derived from self-generated labels.

There is a limited body of work dedicated to understanding GSSL methods. In a notable study, Akhondzadeh et al. [95] explored the use of probing tasks to measure and contrast the richness of graph representations derived from various models. A significant revelation from this study is that transformer-based GNNs capture chemically pertinent information more effectively than message-passing GNNs. Further integrating the data from 3D structures, Liu et al. [96] presented Geom3D — a comprehensive framework for benchmarking geometric representation learning techniques applicable to molecules, proteins, and materials. This framework encompasses 16 cutting-edge geometric models and evaluates their efficacy across 46 diverse scientific challenges, spanning small molecules, proteins, and crystalline substances. An innovative aspect of Geom3D is its approach to categorising geometric models into three groups: invariant, spherical frame equivariant, and vector frame equivariant.

Table 7: Hyperparameters search space of GSSL .

| METHOD | HYPERPARAMETERS | # MODELS |
|---|---|---|
| EDGEPRED | LEARNING RATE | 15 |
| ATTRMASK | MASK RATE, LEARNING RATE | 300 |
| GPT-GNN | LEARNING RATE | 15 |
| INFOGRAPH | LEARNING RATE | 15 |
| GROVER | LEARNING RATE, "CONTEXTURAL" OR "MOTIF"-BASED LOSS | 30 |
| CONT.PRED | LEARNING RATE, CONTEXT SIZE, # NEGATIVE SAMPLES | 300 |
| GRAPHCL | LEARNING RATE, AUG STRENGTH, AUG PROB | 360 |
| JOAO | LEARNING RATE, GAMMA, LOSS VERSION (V1 OR V2) | 300 |
| GRAPHMVP | LEARNING RATE, TEMPERATURE, ALPHA2, # CONFORMER | 540 |
| Total | | **1875** |

## B.3 Hyperparameters search

We search the optimal hyperparameters of pre-training methods, details are summarised in Tables 7 to 9. We select the best hyperparameter of each GSSL method based on their averaged score on downstream datasets (in Table 1, linear models).

We provide details in calculating the **1875** GNN configurations in Table 7. The number of probe models is calculated as follows: 1875 * 8 (MPP datasets) * 2 (Fix, FT) * 3 (Seed) + 9 (GSSL, Optimal) * 10 (Topological Metrics) * 3 (Seed) + 9 (GSSL, Optimal) * 24 (Substructure) * 3 (Seed) = **90918**. It takes over 4 terabytes to save these pre-trained models.

## B.4 Probe models

Ideally, probe models should be neither too simple to capture the representation's information, nor too powerful to learn precise property prediction themselves. If overly powerful, the probe's performance might not accurately reflect the information embedded in the representations. **In light of these complexities, we select a linear model as our probe, aligning with the choice prevalent in most probe studies.**

Table 8: Range of Grid Search on Hyperparameters Space.

| HYPERPARAMETERS | RANGE |
|---|---|
| LEARNING RATE, ALL BUT GRAPHMVP | [0.01, 0.005, 0.001, 0.0005, 0.0001] |
| LEARNING RATE, GRAPHMVP | [0.001, 0.0005, 0.0001] |
| MASK RATE | [0.05, 0.10, ..., 0.95] |
| CONTEXT SIZE | [2, 3, 4, 5] |
| # NEGATIVE SAMPLES | [1, 2, 3, 4, 5] |
| AUG STRENGTH | [0.2, 0.4, 0.6, 0.8] |
| AUG PROBABILITY | [0.1, 0.2, ..., 1.0] |
| GAMMA | [0.1, 0.2, ..., 1.0] |
| TEMPERATURE | [0.1, 0.2, 0.5, 1, 2] |
| ALPHA2 | [0.1, 1, 10] |
| # CONFORMER | [1, 5, 10, 20] |

Table 9: Optimal hyperparameters based on **linear probing and finetuning scores** on MPP tasks.

| METHOD | OPTIMAL HYPERPARAMETERS (left: probing / right: fine-tuning) |
|---|---|
| EDGEPRED | LEARNING RATE=1e-2/1e-2 |
| ATTRMASK | LEARNING RATE=1e-4/5e-4, MASK RATE=0.85/0.50 |
| GPT-GNN | LEARNING RATE=1e-2/1e-4 |
| INFOGRAPH | LEARNING RATE=1e-4/1e-4 |
| GROVER | LEARNING RATE=1e-4/1e-3, "MOTIF"/"CONTEXTUAL"-BASED LOSS |
| CONT.PRED | LEARNING RATE=1e-3/5e-3, CONTEXT SIZE=1/1, # NEGATIVE SAMPLES=5/1 |
| GRAPHCL | LEARNING RATE=1e-3/1e-3, AUG STRENGTH=0.2/0.6, AUG PROB=0.8/0.5, |
| JOAO | LEARNING RATE=1e-3/1e-3, GAMMA=0.9/0.6, "V1"/"V1"-VERSION LOSS |
| GRAPHMVP | LEARNING RATE=5e-4/5e-4, ALPHA2=0.1/10.0, TEMPERATURE=0.1/0.2, # CONFORMER=5/5 |

## B.5 Computation efficiency

We present the number of trainable parameters alongside the average training time per epoch (utilising a single A100 GPU) for each GSSL method in Table 10.

Table 10: Computational efficiency of GSSL methods

| METHOD | NUMBER OF PARAMETERS (Million) | TRAINING TIME (Second) |
|--------|-------------------------------|------------------------|
| EDGEPRED | 7.462 | 101 |
| ATTRMASK | 7.606 | 38 |
| GPT-GNN | 7.606 | 972 |
| INFOGRAPH | 7.823 | 40 |
| GROVER | 7.566 | 39 |
| CONT.PRED | 12.00 | 202 |
| GRAPHCL | 8.186 | 65 |
| JOAO | 8.186 | 382 |
| GRAPHMVP | 15.84 | 119 |

### B.6 Practical guides for future research

In summary, for prospective advancements in Graph SSL, the following practical guidelines should be considered:

- Develop novel pretext tasks that emphasise beneficial invariances and geometry, as unveiled by MOLGRAPHEVAL probes. This includes tasks that enhance substructure modelling and preserve local topology.

- Relying solely on either probing or fine-tuning may not provide a comprehensive understanding. It's noteworthy that weak probing performance doesn't necessarily correlate with subpar fine-tuning outcomes (*e.g.*,, GRAPHMVP). The method of choice should be based on the specific downstream task, whether it's property prediction, generation, optimisation, or interaction modelling.

- Innovate superior data augmentation techniques specifically for molecular graphs to produce valuable views for contrastive learning. Probes can serve as an instrumental means to assess the quality of these augmentations.

- The prevailing notion of achieving a more uniform embedding space is helpful doesn't always hold true in the context of molecular graphs.

- Additionally, probes can be harnessed as a potent tool to scrutinise the impacts of varied negative sampling and augmentation strategies, exemplified by the comparison between GRAPHCL and JOAO.

## C Randomised embeddings

**How GNN models are initialised.** We first analyse how weights in the GNNs are initialised (PyTorch and PyG).

- **Edge Embedding Layers** uses 'xavier uniform', essentially samples from uniform distribution

- **GNN Layers** in fact only have MLP weights (see PyG Doc), same initialisation as Linear layers.

- **Linear Layers** samples from uniform distribution for both weight and bias (PyTorch Doc)

Since all the weights (Edge embedding layers, GNN layers, and Linear layers) in the GINs are extracted from some uniform distribution of some positive ranges. As the GIN layer essentially consists of multiplications and additions, the expected statistics of the node embeddings from randomised GINs are proportional to the number of connected neighbours (i.e., node degrees). Therefore, the randomised embeddings form discriminative clusters in Fig. 4. As for other node-level metrics (in Fig. 8), we don't observe a good clustering formed from randomised embeddings.

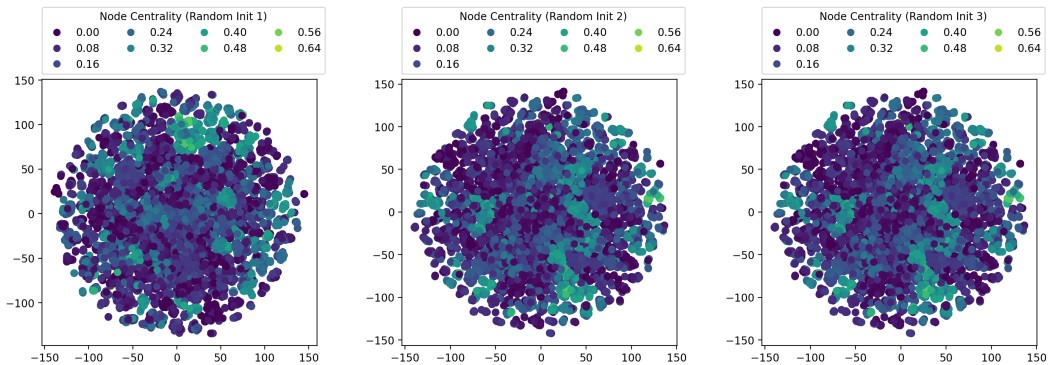

Figure 8: T-SNE visualisation of node centrality produced by RANDOM (three different seeds).

# D Substructure

## D.1 Discussions on substructure counting

A recent study [97] asserts that message-passing neural networks (MPNNs), including Graph Isomorphism Networks (GIN), struggle with the exact counting of certain induced-subgraph structures. While this observation does not directly indicate that Graph Self-Supervised Learning (GSSL) aids in precise subgraph counting, we find that the knowledge acquired from GSSL pre-training can be construed as substructure awareness.

Conversely, another study [98] demonstrates that in the context of molecular graphs, the theoretical expressiveness limitations of MPNNs, as described by the Weisfeiler-Lehman Isomorphism test, do not impair generalisation performance in real-world datasets. As evidenced in Table .6, GIN models, especially those that are pre-trained, exhibit substantial proficiency in identifying substructures.

In conclusion, the question of whether the expressiveness of the backbone model is a limiting factor in molecular domains remains a topic of ongoing debate. The investigation of the backbone model operates independently of the GSSL pretraining analysis undertaken in our study. As such, we propose deferring this line of inquiry for future exploration.

## D.2 Detailed description and performance

In Table 11, we provide the descriptions of the molecular substructures (mainly from documents on the rdkit.Chem.Fragments, textbooks [99] and Wikipedia). We also listed some molecular properties that are affected by these substructures. Table 12 and Table 13 report the detailed performance of substructure property prediction.

Table 11: Descriptions of substructure and which properties their existence would affect. Molecules containing X substructure are usually named as "X-Compounds", "X derivatives" and "X moieties".

| Substructure | Description & Affected molecular properties |
|---|---|
| allylic | Allylic oxidations have featured in hundreds of chemical syntheses, due to their particular electrochemical properties [100–102]. |
| amide | An amidine is a compound with the general formula RC(=O)NR'R", where R, R', and R" represent organic groups or hydrogen atoms. It has significant impacts on the mechanical, acid-base, and solubility properties of molecules [wikipage]. |

| | |
|---|---|
| amidine | Amidines are organic compounds with the functional group RC(NR)NR2, where the R groups can be the same or different. They are the imine derivatives of amides (RC(O)NR2). Amidines are much more basic than amides and are among the strongest uncharged/unionized bases.

Several drug or drug candidates feature amidine substituents. Examples include the antiprotozoal Imidocarb, the insecticide amitraz , the anthelmintic tribendimidine, and xylamidine, an antagonist at the 5HT2A receptor [wikipage]. |
| Azo | Azo compounds are compounds bearing the functional group diazenyl R-N=N-R', in which R and R' can be either aryl or alkyl. Certain azo compounds are known to have antibiotic, antiviral, antifungal, antineoplastic, and cytotoxic properties [103]. |
| benzene | Benzene (aromatic rings) is an organic chemical compound with the molecular formula C6H6. Aromatic rings are important residues for biological interactions and appear to a large extent as part of protein–drug and protein–protein interactions. They are relevant for both protein stability and molecular recognition processes due to their natural occurrence in aromatic aminoacids (Trp, Phe, Tyr and His) as well as in designed drugs since they are believed to contribute to optimising both affinity and specificity of drug-like molecules [104]. |
| epoxide | An epoxide is a cyclic ether with a three-atom ring. Epoxide-containing molecules have therapeutic value. The main therapeutic interest is as anticancer agents. The main mechanisms are enzyme inhibition, induction of cell cycle arrest, apoptosis. Other therapeutic interests are for heart failure, infections, gastrointestinal diseases [105]. |
| ether | Ethers are a class of organic compounds that contain an ether group-an oxygen atom connected to two alkyl or aryl groups. They have the general formula R-O-R', where R and R' represent the alkyl or aryl groups. The C-O bonds that comprise simple ethers are strong. They are unreactive toward all but the strongest bases. Although generally of low chemical reactivity, they are more reactive than alkanes. Some important reactions include cleavage, peroxide formation, lewis bases, and alpha-halogenation[wikipage]. |
| furan | Furan is a heterocyclic organic compound, consisting of a five-membered aromatic ring with four carbon atoms and one oxygen atom. The furan ring present in the chemical structures may be one of the domineering factors to bring about the toxic response resulting from the generation of reactive epoxide or cis-enedial intermediates which are of the potential to react with biomacromolecules [106, 107]. |
| guanido | Guanidine is the compound with the formula HNC(NH2)2. It is a colourless solid that dissolves in polar solvents. It is a strong base that is used in the production of plastics and explosives. Most guanidine derivatives are in fact salts containing the conjugate acid[wikipage].

Guanidine-containing derivatives constitute a very important class of therapeutic agents suitable for the treatment of a wide spectrum of diseases [108]. |
| halogen | The halogens are a group in the periodic table consisting of five or six chemically related elements: fluorine (F), chlorine (Cl), bromine (Br), iodine (I), and astatine (At). Halogens are highly reactive, and as such can be harmful or lethal to biological organisms in sufficient quantities. This high reactivity is due to the high electronegativity of the atoms due to their high effective nuclear charge[wikipage].

A significant number of drugs and drug candidates in clinical development are halogenated structures. For a long time, insertion of halogen atoms on hit or lead compounds was predominantly performed to exploit their steric effects, through the ability of these bulk atoms to occupy the binding site of molecular targets [109]. |

| | |
|---|---|
| imidazole | Imidazole is an organic compound with the formula C3N2H4. It is a white or colourless solid that is soluble in water, producing a mildly alkaline solution. This ring system is present in important biological building blocks, such as histidine and the related hormone histamine. Many drugs contain an imidazole ring, such as certain antifungal drugs, the nitroimidazole series of antibiotics, and the sedative midazolam[wikipage]. |
| imide | In organic chemistry, an imide is a functional group consisting of two acyl groups bound to nitrogen. Being highly polar, imides exhibit good solubility in polar media. The N–H center for imides derived from ammonia is acidic and can participate in hydrogen bonding. Unlike the structurally related acid anhydrides, they resist hydrolysis and some can even be recrystallized from boiling water[wikipage]. Immunomodulatory imide drugs (IMiDs) are a class of immunomodulatory drugs (drugs that adjust immune responses) containing an imide group. |
| lactam | A beta-lactam ($\beta$-lactam) ring is a four-membered lactam. The $\beta$-lactam ring is part of the core structure of several antibiotic families, the principal ones being the penicillins, cephalosporins, carbapenems, and monobactams, which are, therefore, also called $\beta$-lactam antibiotics [wikipage]. |
| morpholine | Morpholine is an organic chemical compound having the chemical formula O(CH2CH2)2NH. Morpholine is a heterocycle featured in numerous approved and experimental drugs as well as bioactive molecules. It is often employed in the field of medicinal chemistry for its advantageous physicochemical, biological, and metabolic properties, as well as its facile synthetic routes [110]. |
| NO (hydroxylamine) | Hydroxylamine is an organic compound with the formula NH 2OH. The material is a white crystalline, hygroscopic compound. Hydroxylamines and their derivatives are powerful aminating reagents, which are often used for arene C–H and X-H aminations (X=O, N, S, P) as well as Schmidt-type reaction46, serving as alternative ways to introduce amino groups on various chemical skeletons [111]. |
| oxazole | Oxazoles is a doubly unsaturated 5-membered ring having one oxygen atom at position 1 and a nitrogen at position 3 separated by a carbon in-between. Substitution pattern in oxazole derivatives play a pivotal role in delineating the biological activities like antimicrobial, anticancer, antitubercular anti-inflammatory, antidiabetic, antiobesity and antioxidant etc [112]. |
| piperdine | Piperidine is an organic compound with the molecular formula (CH2)5NH. This heterocyclic amine consists of a six-membered ring containing five methylene bridges (–CH2–) and one amine bridge (–NH–). Piperidine and its derivatives are ubiquitous building blocks in pharmaceuticals[26] and fine chemicals. The piperidine structure is found in, for example: Icaridin, SSRIs, stumulants and nootropics, SERM etc. Piperidine is also commonly used in chemical degradation reactions, such as the sequencing of DNA in the cleavage of particular modified nucleotides. Piperidine is also commonly used as a base for the deprotection of Fmoc-amino acids used in solid-phase peptide synthesis [wikipage]. |
| piperazine | Piperazine is an organic compound that consists of a six-membered ring containing two nitrogen atoms at opposite positions in the ring. Many currently notable drugs contain a piperazine ring as part of their molecular structure ("Substituted piperazine"). Examples include: Antianginals, Antidepressants, Antihistamines etc [wikipage]. |
| pyridine | Pyridine is a basic heterocyclic organic compound with the chemical formula C5H5N. Pyridine moieties are often used in drugs because of their characteristics such as basicity, water solubility, stability, and hydrogen bond-forming ability, and their small molecular size [113]. Pyridine-based ring systems are one of the most extensively used heterocycles in the field of drug design, primarily due to their profound effect on pharmacological activity, which has led to the discovery of numerous broad-spectrum therapeutic agents [114]. |

| tetrazole | Tetrazoles are a class of synthetic organic heterocyclic compound, consisting of a 5-member ring of four nitrogen atoms and one carbon atom. Tetrazole derivatives are a prime class of heterocycles, very important to medicinal chemistry and drug design due to not only their bioisosterism to carboxylic acid and amide moieties but also to their metabolic stability and other beneficial physicochemical properties [115]. |
|---|---|
| thiazole | Thiazole, or 1,3-thiazole, is a heterocyclic compound that contains both sulfur and nitrogen. The versatility of thiazole nucleus demonstrated by the fact that it is an essential part of penicillin nucleus and some of its derivatives which have shown antimicrobial (sulfazole), antiretroviral (ritonavir), antifungal (abafungin), antihistaminic and antithyroid activities [116]. |
| thiophene | Thiophene is a heterocyclic compound with the formula C4H4S. In medicine, thiophene derivatives shows antimicrobial, analgesic and anti-inflammatory, antihypertensive, and antitumor activity while they are also used as inhibitors of corrosion of metals or in the fabrication of light-emitting diodes in material science [117]. |
| urea | Urea, also known as carbamide, is an organic compound with chemical formula $CO(NH_2)_2$. This amide has two -$NH_2$ groups joined by a carbonyl (C=O) functional group. It has similar effects as amide groups. |

Table 12: Substructure detection, part I. We **bold** the best and underline the worst scores.

| | allylic | amide | amidine | azo | benzene | epoxide | ether | furan | guanido | halogen | imidazole | imide |
|---|---|---|---|---|---|---|---|---|---|---|---|---|
| RANDOM | 0.959 | 16.917 | 0.054 | 0.020 | 1.100 | 0.024 | 2.024 | 0.036 | 0.126 | 1.127 | 0.080 | 0.062 |
| EDGEPRED | 0.780 | 14.173 | 0.046 | 0.018 | 0.797 | 0.021 | 1.608 | 0.033 | 0.098 | 0.939 | 0.074 | 0.031 |
| ATTRMASK | 0.926 | 14.703 | 0.047 | 0.019 | 0.976 | 0.022 | 1.742 | 0.028 | 0.112 | 0.501 | 0.077 | 0.029 |
| GPT-GNN | 0.872 | 15.629 | 0.044 | 0.017 | 0.783 | 0.021 | 1.912 | 0.023 | 0.117 | 0.341 | 0.077 | 0.037 |
| INFOGRAPH | 0.740 | 6.747 | 0.050 | 0.019 | 0.583 | 0.022 | 1.128 | 0.021 | 0.086 | 0.706 | 0.062 | 0.038 |
| CONT.PRED | 1.040 | 16.636 | 0.053 | 0.020 | 0.980 | 0.023 | 1.787 | 0.034 | 0.126 | 1.075 | 0.078 | 0.033 |
| GROVER | 0.715 | **6.576** | **0.025** | **0.008** | 0.558 | 0.023 | **0.957** | **0.008** | **0.064** | **0.298** | 0.069 | **0.021** |
| GRAPHCL | 0.652 | 7.598 | 0.039 | 0.016 | **0.525** | 0.023 | 1.077 | 0.012 | 0.080 | 0.319 | 0.051 | 0.026 |
| JOAO | **0.654** | 7.926 | 0.043 | 0.015 | 0.531 | 0.023 | 1.071 | 0.013 | 0.085 | 0.310 | **0.048** | 0.026 |
| GRAPHMVP | 0.905 | 6.992 | 0.043 | 0.017 | 0.649 | **0.019** | 1.037 | 0.019 | 0.084 | 0.311 | 0.060 | 0.027 |
| SSL Worse (#) | 1 | 0 | 0 | 0 | 0 | 0 | 0 | 0 | 0 | 0 | 0 | 0 |

Table 13: Substructure detection, part II. We **bold** the best and underline the worst scores.

| | lactam | morpholine | NO | oxazole | piperdine | piperzine | pyridine | tetrazole | thiazole | thiophene | urea |
|---|---|---|---|---|---|---|---|---|---|---|---|
| RANDOM | 0.018 | 0.031 | 0.022 | 0.009 | 0.212 | 0.058 | 0.176 | 0.014 | 0.040 | 0.052 | 0.045 |
| EDGEPRED | 0.016 | 0.020 | 0.022 | 0.008 | 0.182 | 0.048 | 0.158 | 0.014 | 0.039 | 0.046 | 0.041 |
| ATTRMASK | 0.016 | 0.028 | 0.022 | 0.008 | 0.192 | 0.053 | 0.174 | 0.014 | 0.038 | 0.044 | 0.044 |
| GPT-GNN | 0.012 | 0.021 | 0.022 | 0.008 | 0.177 | 0.049 | 0.128 | 0.014 | 0.039 | 0.041 | 0.037 |
| INFOGRAPH | 0.012 | 0.026 | 0.024 | 0.007 | 0.185 | 0.048 | 0.127 | 0.016 | 0.029 | 0.036 | 0.046 |
| CONT.PRED | 0.016 | 0.031 | 0.022 | 0.008 | 0.214 | 0.059 | 0.170 | 0.014 | 0.039 | 0.047 | 0.045 |
| GROVER | 0.014 | 0.022 | 0.028 | **0.004** | 0.158 | 0.043 | 0.096 | **0.005** | **0.018** | **0.018** | **0.014** |
| GRAPHCL | 0.014 | **0.019** | 0.021 | 0.006 | 0.169 | **0.031** | 0.084 | 0.009 | 0.023 | **0.018** | 0.023 |
| JOAO | 0.014 | 0.021 | **0.020** | 0.006 | 0.168 | 0.035 | **0.082** | 0.009 | 0.022 | **0.018** | 0.025 |
| GRAPHMVP | **0.011** | 0.021 | 0.022 | 0.006 | **0.155** | 0.038 | 0.129 | 0.009 | 0.027 | 0.025 | 0.028 |
| SSL Worse (#) | 0 | 0 | 2 | 0 | 1 | 1 | 0 | 1 | 0 | 0 | 1 |

## D.3 Cramer's V

**Cramér's V** quantifies the strength of the association between the molecular substructure counts (*i.e.*,, chemical fragments) and their biochemical properties. It is defined as:

$$V = \sqrt{\chi^2 / (n \cdot \min(k - 1, r - 1))} = \sqrt{\chi^2/n} \quad (r \equiv 2) \tag{5}$$

where $n$ is the sample size, $k$ and $r$ are the total number of substructure counts and property categories (binary), respectively. The Chi-squared statistics $\chi^2$ is then calculated as:

$$\chi^2 = \sum_{i,j} \left( n_{(i,j)} - n_{(i,\cdot)} \cdot n_{(\cdot,j)}/n \right)^2 \Big/ \left( n_{(i,\cdot)} \cdot n_{(\cdot,j)}/n \right) \tag{6}$$

where $n_{(i,j)}$ is the total occurrence for the pair of $(i, j)$. Here $i$ is the specific count of a certain substructure, and $j$ represents the certain outcome of a molecular biochemical property. Cramér's V value ranges from 0 to 1, representing the associated strength between two categorical variables.

Table 14: Cramér's V between molecular substructure counts and downstream properties.

| Pre-training | BBBP | Tox21 | ToxCast | Sider | ClinTox | MUV | HIV | Bace | Avg (Task) | Avg (Data) |
|---|---|---|---|---|---|---|---|---|---|---|
| allylic | 0.1602 | 0.1345 | 0.1156 | 0.1276 | 0.0935 | 0.0413 | 0.0280 | 0.1186 | 0.1144 | 0.1024 |
| amide | 0.2692 | 0.0490 | 0.0858 | 0.1841 | 0.1326 | 0.0235 | 0.0689 | 0.2556 | 0.0881 | 0.1336 |
| amidine | 0.0360 | 0.0291 | 0.0412 | 0.0323 | 0.0158 | 0.0117 | 0.0396 | 0.1328 | 0.0399 | 0.0423 |
| azo | 0.0400 | 0.0399 | 0.0393 | 0.0277 | 0.0123 | 0.0007 | 0.2082 | - | 0.0384 | 0.0526 |
| benzene | 0.1476 | 0.1632 | 0.1691 | 0.1149 | 0.1112 | 0.0289 | 0.1374 | 0.1091 | 0.1630 | 0.1227 |
| epoxide | 0.0273 | 0.0481 | 0.0449 | 0.0300 | 0.0049 | 0.0005 | 0.0086 | - | 0.0437 | 0.0235 |
| ether | 0.2314 | 0.0694 | 0.1060 | 0.1069 | 0.1023 | 0.0185 | 0.0498 | 0.1821 | 0.1034 | 0.1083 |
| furan | 0.0635 | 0.0257 | 0.0387 | 0.0227 | 0.0061 | 0.0311 | 0.0148 | 0.0135 | 0.0375 | 0.0270 |
| guanido | 0.0765 | 0.0201 | 0.0509 | 0.0715 | 0.0286 | 0.0057 | 0.0094 | 0.1088 | 0.0499 | 0.0464 |
| halogen | 0.1488 | 0.0849 | 0.1827 | 0.0773 | 0.0908 | 0.0143 | 0.0347 | 0.2353 | 0.1721 | 0.1086 |
| imidazole | 0.0601 | 0.0427 | 0.0492 | 0.0460 | 0.1212 | 0.0102 | 0.0398 | 0.1280 | 0.0483 | 0.0622 |
| imide | 0.0951 | 0.0246 | 0.0401 | 0.0428 | 0.0518 | 0.0094 | 0.0188 | - | 0.0392 | 0.0404 |
| lactam | 0.4263 | 0.0184 | 0.0116 | 0.0646 | 0.0543 | 0.0006 | 0.0048 | - | 0.0182 | 0.0830 |
| morpholine | 0.0512 | 0.0126 | 0.0343 | 0.0268 | 0.0425 | 0.0068 | 0.0101 | 0.0668 | 0.0329 | 0.0314 |
| N_O | 0.0438 | 0.0195 | 0.0467 | 0.0391 | 0.0709 | 0.0195 | 0.0144 | 0.0537 | 0.0452 | 0.0385 |
| oxazole | 0.0126 | 0.0184 | 0.0321 | 0.0359 | 0.0123 | 0.0079 | 0.0080 | 0.0364 | 0.0312 | 0.0205 |
| piperdine | 0.1450 | 0.0305 | 0.0844 | 0.0575 | 0.0418 | 0.0079 | 0.0226 | 0.0935 | 0.0803 | 0.0604 |
| piperzine | 0.0509 | 0.0214 | 0.0421 | 0.0776 | 0.0648 | 0.0111 | 0.0192 | 0.0063 | 0.0424 | 0.0367 |
| pyridine | 0.0598 | 0.0402 | 0.0549 | 0.0338 | 0.0833 | 0.0129 | 0.0300 | 0.1747 | 0.0529 | 0.0612 |
| tetrazole | 0.1161 | 0.0158 | 0.0251 | 0.0300 | 0.0286 | 0.0083 | 0.0123 | 0.0334 | 0.0247 | 0.0337 |
| thiazole | 0.1389 | 0.0521 | 0.0345 | 0.0445 | 0.0183 | 0.0118 | 0.0173 | 0.0539 | 0.0348 | 0.0464 |
| thiophene | 0.0356 | 0.0467 | 0.0472 | 0.0315 | 0.0113 | 0.0166 | 0.0081 | 0.0438 | 0.0456 | 0.0301 |
| urea | 0.0790 | 0.0236 | 0.0506 | 0.0471 | 0.0268 | 0.0079 | 0.0329 | 0.0516 | 0.0489 | 0.0399 |

## D.4 Distribution

We plot the distribution of 24 molecular substructures, the y-axis (*i.e.*, counts) is log scale. Alongside each substructure name, we includes its average counts (*i.e.*,, ground truth). While certain substructures, such as amidine and azo, are infrequently found in the molecules from the eight downstream datasets, we note that others like Amide and Benzene are prevalent. These substructures are recognised for their significant relevance to molecular properties, as detailed in Table 4.

**Allylic Oxide**.

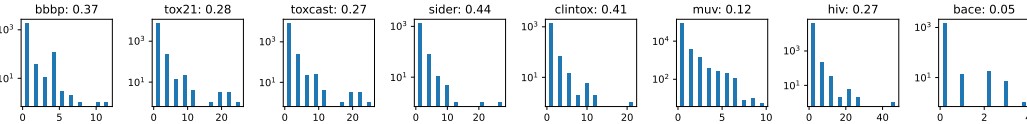

**Amide**.

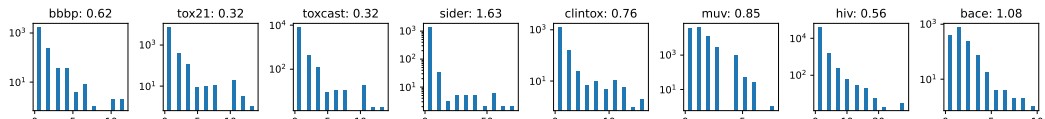

**Amidine**.

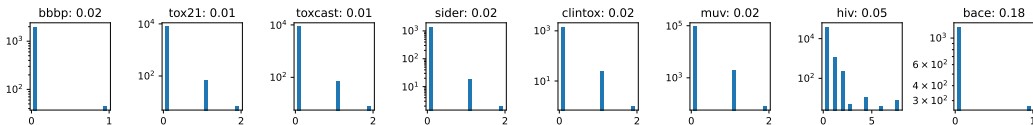

**AZO**.

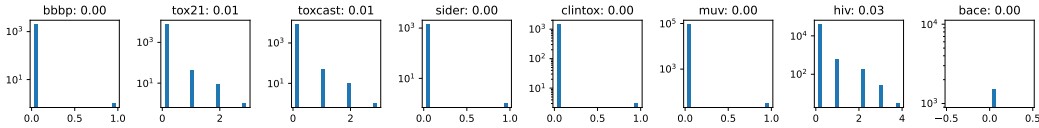

**Benzene**.

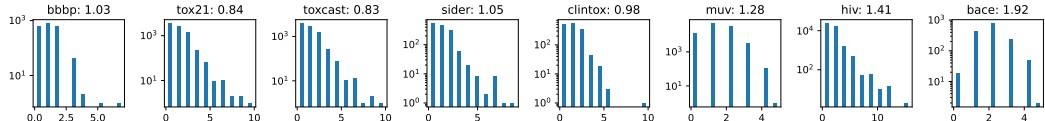

**Epoxide**.

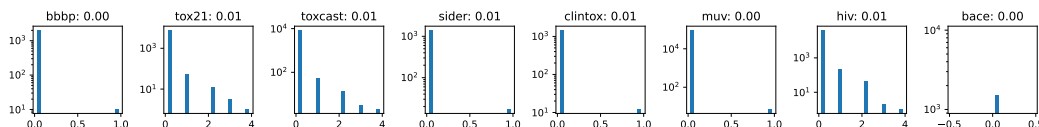

**Ether**.

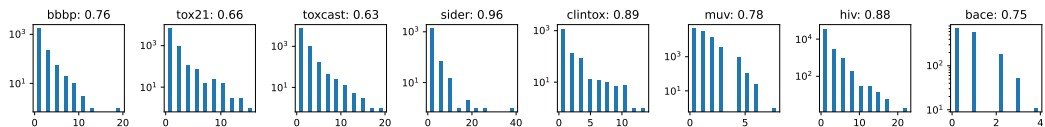

**Furan**.

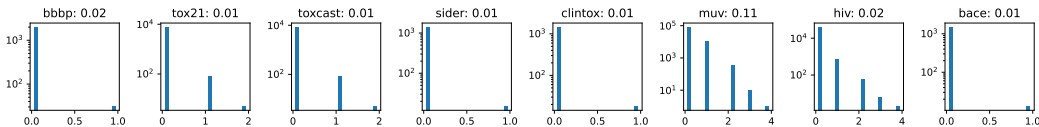

**Guanido**.

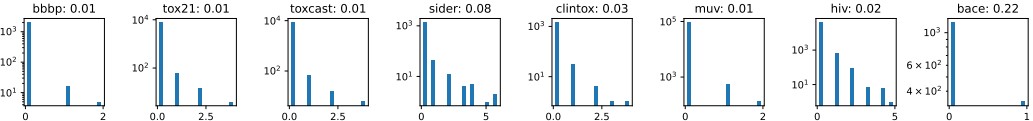

**Halogen**.

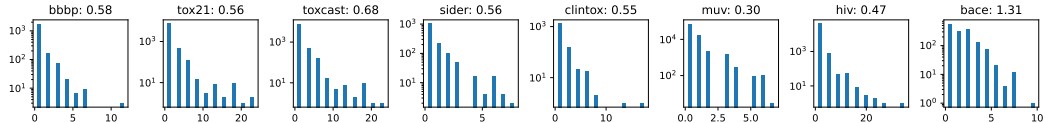

**Imidazole**.

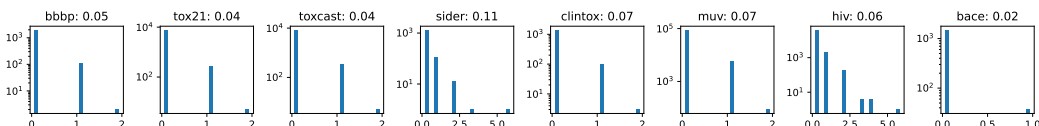

**Imide**.

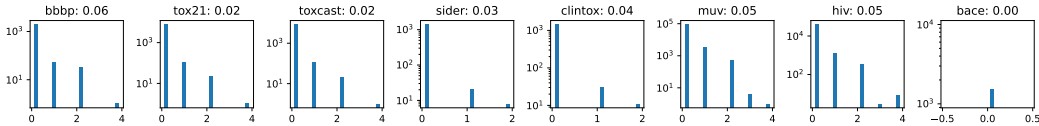

**Lactam**.

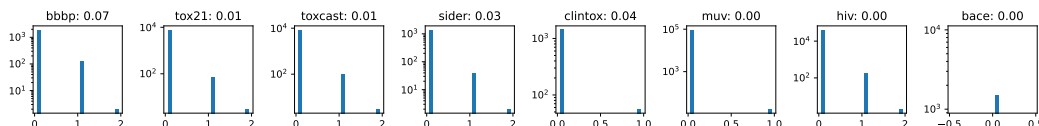

**Morpholine**.

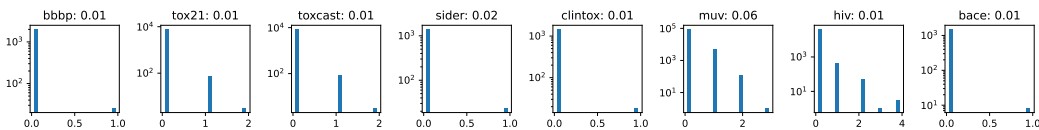

**N_O**.

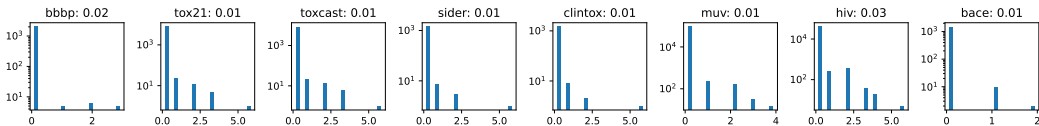

**Oxazole**.

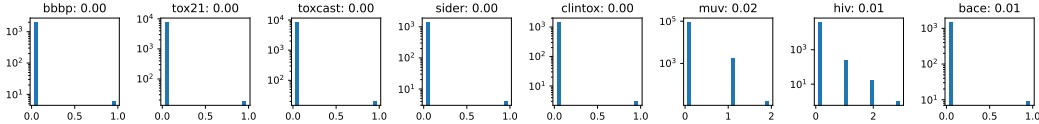

**Piperdine**.

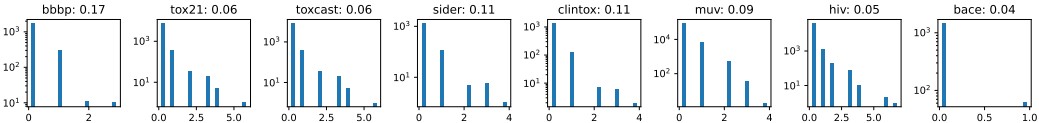

**Piperzine**.

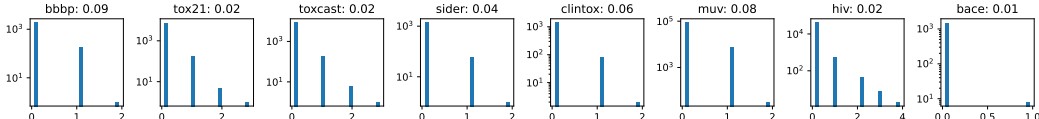

**Pyridine**.

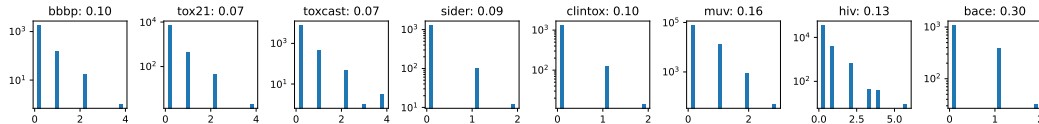

**Tetrazole**.

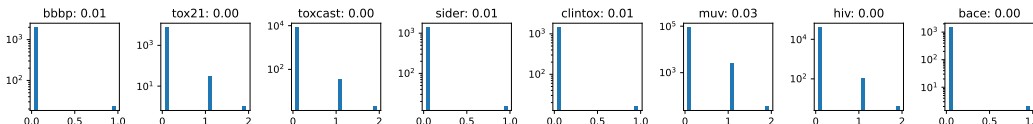

**Thiazole**.

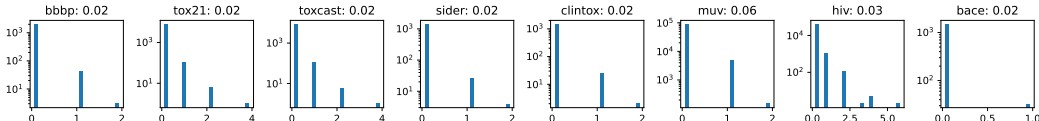

**Thiophene**.

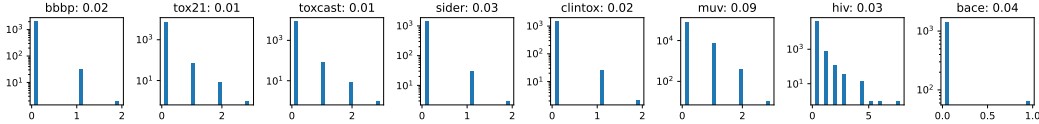

**Urea**.

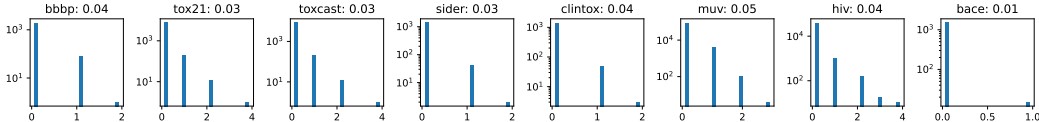

# E    More results on metrics

## E.1    Topological metric

We have depicted the distribution of structural metrics and substructures with respect to the downstream datasets using histograms. Please note that the vertical axes may occasionally be represented on a logarithmic scale.

**Node Degree**.

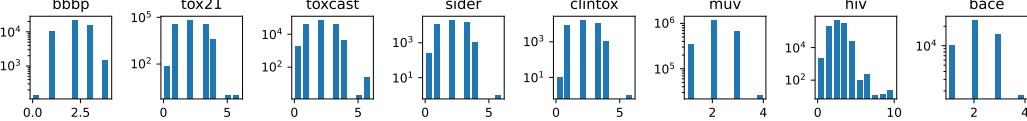

**Node Centrality**.

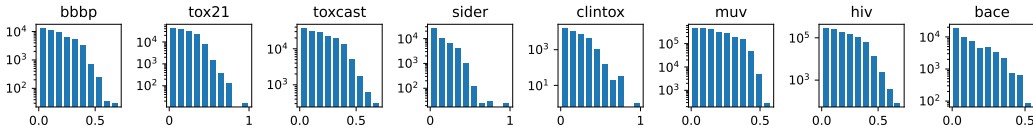

**Node Clustering Coefficient**.

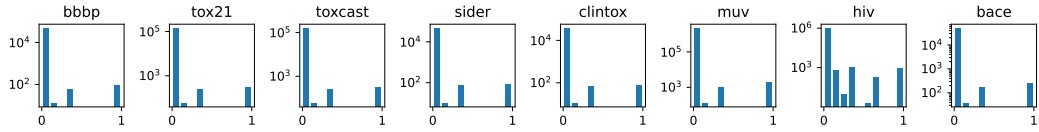

**Link Prediction**.

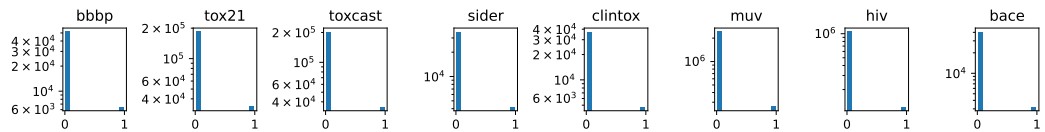

**Jaccard Coefficient**.

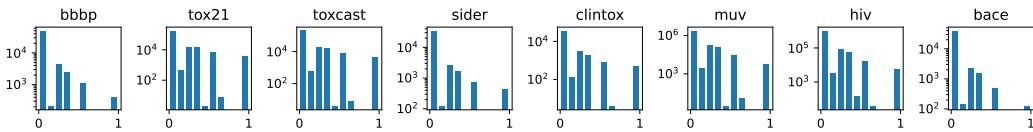

**Katz Index**.

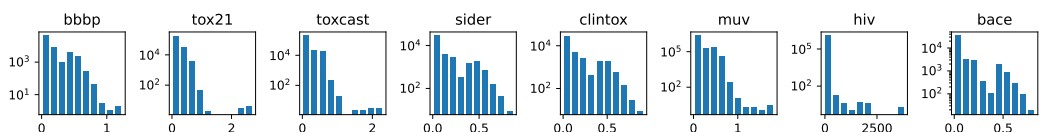

**Cycle Basis**.

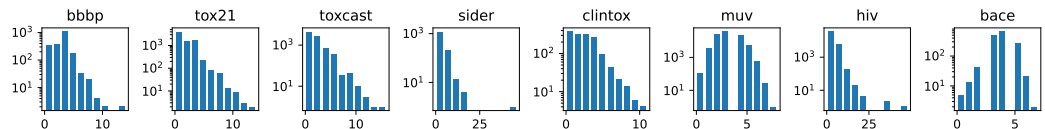

**Graph Diameter**.

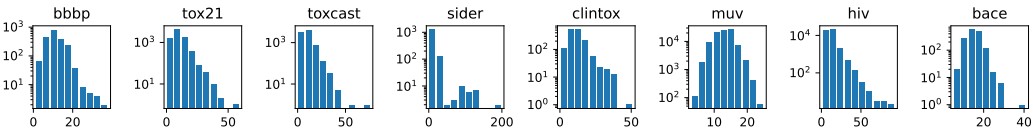

**Aassortativity Coefficient**.

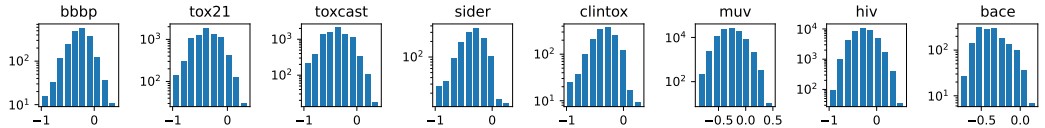

**Average Clustering Coefficient**.

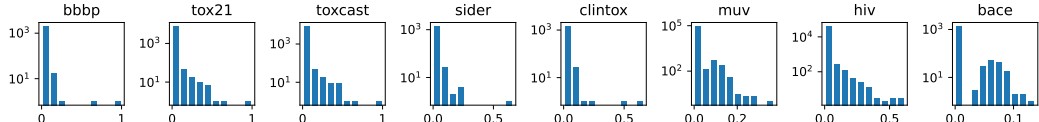

## E.2 Spectrum

Our observations suggest a positive correlation between the magnitudes of singular values in pre-trained embedding spaces and their performance in the downstream MPP tasks.

Table 15: Benchmarking the magnitude of the singular values (node/graph) in the spectrum, BBBP.

| Dimension | 1 | 50 | 100 | 200 | 300 |
|---|---|---|---|---|---|
| RANDOM | 1.96e-01/4.81e-02 | 8.14e-06/3.35e-07 | 1.08e-06/3.87e-08 | 9.10e-08/2.20e-09 | 2.44e-09/4.08e-11 |
| EDGEPRED | 1.25e+00/2.12e-01 | 5.30e-03/5.96e-04 | 3.92e-04/3.56e-05 | 1.31e-05/7.10e-07 | 1.97e-07/6.79e-09 |
| ATTRMASK | 9.97e+01/7.57e+00 | 9.71e-03/1.16e-03 | 2.06e-03/2.02e-04 | 2.95e-04/1.96e-05 | 1.86e-05/9.00e-07 |
| GPT-GNN | 7.78e+01/2.46e+01 | 4.21e-03/6.04e-04 | 6.26e-04/7.11e-05 | 5.32e-05/4.40e-06 | 1.29e-06/6.50e-08 |
| INFOGRAPH | 1.29e+02/6.56e+01 | 9.52e-01/3.23e-01 | 2.68e-01/6.54e-02 | 2.43e-02/3.38e-03 | 2.80e-05/2.69e-06 |
| CONT.PRED | 1.81e+01/5.89e+00 | 3.40e-04/3.72e-05 | 4.36e-05/3.42e-06 | 1.65e-06/9.08e-08 | 1.14e-08/3.92e-10 |
| GROVER | 2.21e+02/9.86e+01 | 2.18e+00/5.11e-01 | 1.13e-01/2.04e-02 | 1.68e-02/1.97e-03 | 1.51e-03/1.36e-04 |
| GRAPHCL | 7.63e+01/3.40e+01 | 4.26e-01/7.27e-02 | 1.45e-02/1.76e-03 | 6.84e-04/5.11e-05 | 2.42e-05/1.37e-06 |
| JOAO | 8.12e+01/3.58e+01 | 4.16e-01/7.89e-02 | 2.18e-02/2.67e-03 | 8.63e-04/6.74e-05 | 2.64e-05/1.52e-06 |
| GRAPHMVP | 2.07e+01/1.17e+01 | 2.57e-01/8.22e-02 | 1.94e-02/4.00e-03 | 4.39e-05/5.75e-06 | 1.76e-06/1.71e-07 |
| Correlation | 0.661/0.781 | 0.806/0.927 | 0.879/0.903 | 0.697/0.770 | 0.794/0.794 |

We provide more visualisations of the spectrum of the embedding space from different datasets.

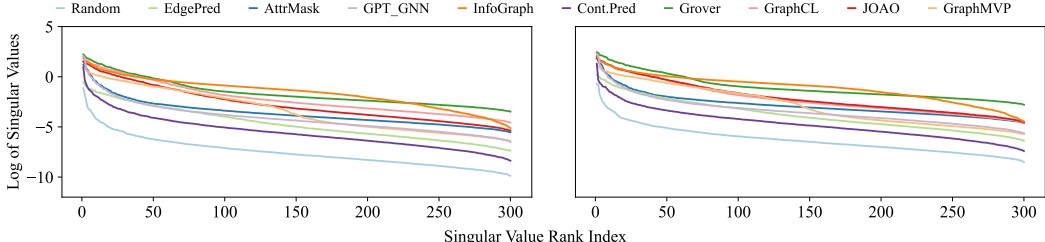

Figure 9: Spectrum of the GSSL embedding space on Tox21 dataset, Left: Node; Right: Graph.

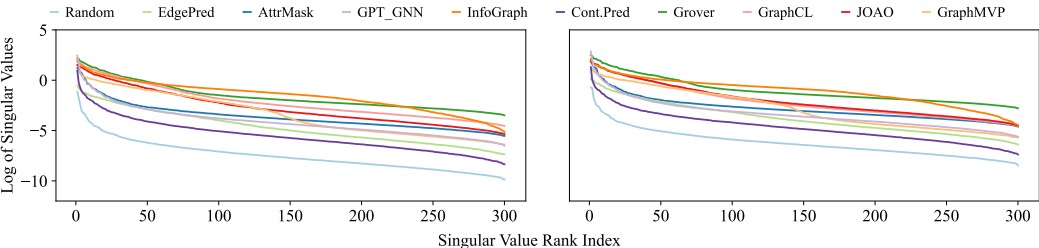

Figure 10: Spectrum of the GSSL embedding space on Toxcast dataset, Left: Node; Right: Graph.

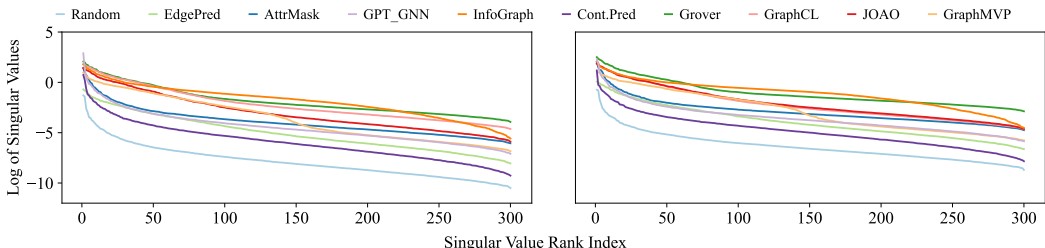

Figure 11: Spectrum of the GSSL embedding space on Sider dataset, Left: Node; Right: Graph.

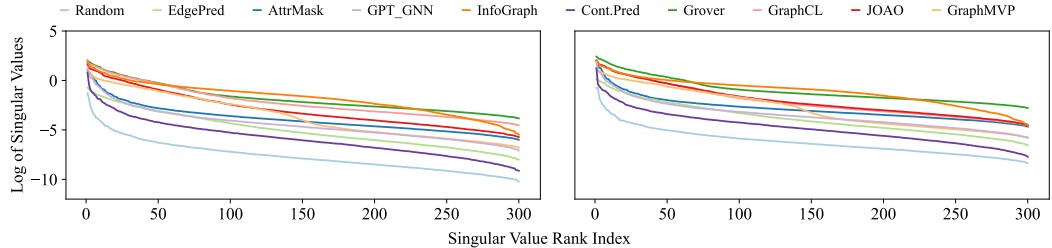

Figure 12: Spectrum of the GSSL embedding space on Clintox dataset, Left: Node; Right: Graph.

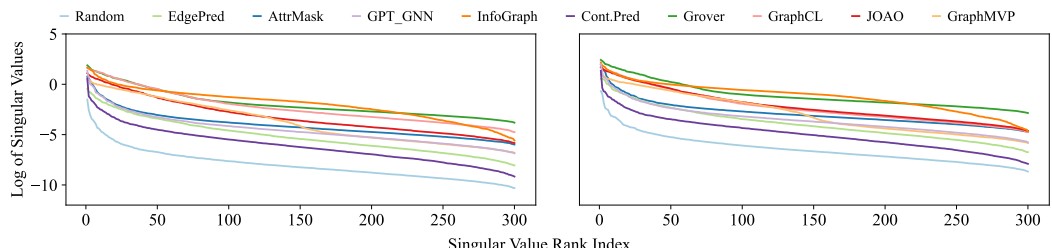

Figure 13: Spectrum of the GSSL embedding space on MUV dataset, Left: Node; Right: Graph.

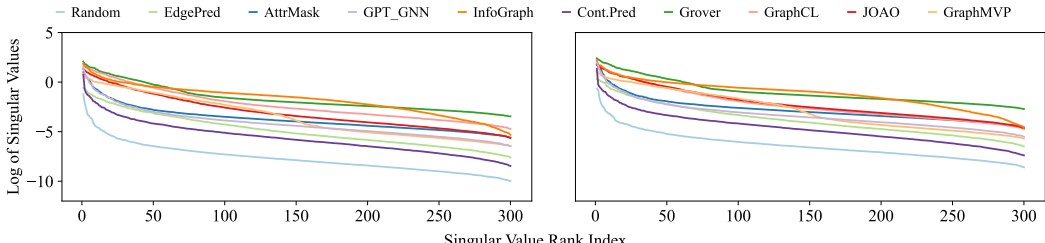

Figure 14: Spectrum of the GSSL embedding space on HIV dataset, Left: Node; Right: Graph.

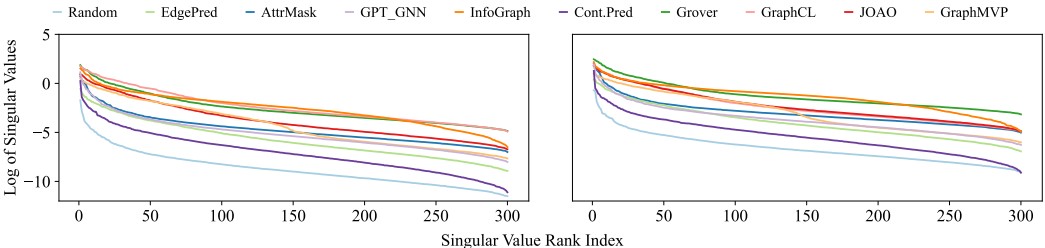

Figure 15: Spectrum of the GSSL embedding space on BACE dataset, Left: Node; Right: Graph.

### E.3 Uniformity

We provide more comprehensive results on the uniformity metric of the embedding space in Table 16. The upper bound for uniformity is zero, while the lower bound, as derived from Corollary 1 in the Appendix of [6], is both data and hyperparameter-dependent.

Table 16: **Evaluating GSSL methods on uniformity**.

| Dataset | BBBP | Tox21 | ToxCast | Sider | ClinTox | MUV | HIV | Bace |
|---|---|---|---|---|---|---|---|---|
| RANDOM | -0.214 | -0.315 | -0.313 | -0.282 | -0.242 | -0.128 | -0.245 | -0.079 |
| EDGEPRED | -2.319 | -2.969 | -2.909 | -2.424 | -2.384 | -2.634 | -2.804 | -1.653 |
| ATTRMASK | -8.332 | -9.348 | -9.560 | -7.786 | -8.399 | -9.241 | -9.979 | -6.389 |
| GPT-GNN | -5.714 | -5.701 | -5.773 | -5.550 | -5.708 | -5.279 | -5.068 | -4.977 |
| INFOGRAPH | -10.065 | -10.925 | -11.310 | -7.623 | -9.831 | -15.337 | -13.506 | -9.620 |
| CONT.PRED | -2.636 | -3.074 | -3.209 | -2.748 | -2.780 | -2.390 | -2.865 | -1.708 |
| GROVER | -10.208 | -12.142 | -12.356 | -11.907 | -9.975 | -15.772 | -14.244 | -10.512 |
| GRAPHCL | -10.010 | -11.073 | -11.458 | -10.155 | -9.845 | -13.390 | -12.530 | -8.513 |
| JOAO | -10.000 | -10.955 | -11.360 | -10.012 | -9.846 | -13.450 | -12.505 | -8.523 |
| GRAPHMVP | -8.864 | -9.770 | -9.863 | -8.782 | -8.825 | -12.405 | -11.301 | -7.540 |
| Correlation | 0.842 | -0.600 | 0.097 | -0.210 | 0.309 | 0.169 | 0.821 | 0.285 |
| p-value | 0.002 | 0.067 | 0.789 | 0.559 | 0.384 | 0.641 | 0.004 | 0.425 |

## F  Extending to more GSSL methods and datasets

We demonstrate that the proposed MOLGRAPHEVAL benchmark can readily be extended to incorporate more pre-training methods, such as GRAPHMAE and GRCL, as well as larger datasets, like the 2M ZINC15. It's pertinent to note that the downstream datasets can also be employed for pre-training. Importantly, we observe that thorough optimisation of pre-training hyperparameters can serve as a crucial factor for performance improvements in pre-training, notwithstanding the advancements in the design of pre-training tasks.

Table 17: **Evaluating GSSL methods on molecular property prediction tasks, on 2M ZINC15.** For each downstream dataset, we report the mean and standard deviation of the ROC-AUC scores over three random scaffold splits. The performance scores are based on the fixed pre-trained embeddings with linear probe models, we also report the average ROC-AUC scores with fine-tuned pre-trained GNN on MPP tasks ("Avg (FT)").

| | BBBP | Tox21 | ToxCast | Sider | ClinTox | MUV | HIV | Bace | Avg | Avg (FT) |
|---|---|---|---|---|---|---|---|---|---|---|
| # Molecules | 2,039 | 7,831 | 8,575 | 1,427 | 1,478 | 93,087 | 41,127 | 1,513 | – | – |
| # Tasks | 1 | 12 | 617 | 27 | 2 | 17 | 1 | 1 | – | – |
| RANDOM | 50.7 $_{\pm2.5}$ | 64.9 $_{\pm0.5}$ | 53.2 $_{\pm0.3}$ | 53.2 $_{\pm1.1}$ | 63.1 $_{\pm2.3}$ | 62.1 $_{\pm1.3}$ | 66.1 $_{\pm0.7}$ | 63.4 $_{\pm1.8}$ | 59.60 | 66.16 |
| ATTRMASK | 49.8 $_{\pm0.6}$ | 66.7 $_{\pm0.3}$ | 52.9 $_{\pm0.4}$ | 53.8 $_{\pm1.7}$ | 62.2 $_{\pm2.9}$ | 52.8 $_{\pm1.7}$ | 69.0 $_{\pm1.4}$ | 66.6 $_{\pm4.9}$ | 59.22 | 69.49 |
| GRAPHCL | 64.9 $_{\pm0.5}$ | 70.5 $_{\pm0.6}$ | 56.1 $_{\pm0.2}$ | 58.0 $_{\pm1.4}$ | 63.4 $_{\pm3.1}$ | 61.2 $_{\pm1.6}$ | 75.6 $_{\pm0.9}$ | 70.9 $_{\pm3.8}$ | 65.07 | 70.09 |
| GRAPHMAE | 58.6 $_{\pm2.3}$ | 64.4 $_{\pm0.6}$ | 55.3 $_{\pm0.1}$ | 57.0 $_{\pm1.8}$ | 67.4 $_{\pm1.9}$ | 57.3 $_{\pm1.5}$ | 71.6 $_{\pm1.2}$ | 51.6 $_{\pm3.6}$ | 60.41 | 68.91 |
| GRCL | 61.8 $_{\pm2.1}$ | 67.5 $_{\pm0.4}$ | 54.3 $_{\pm0.4}$ | 55.8 $_{\pm1.3}$ | 62.0 $_{\pm2.4}$ | 61.9 $_{\pm1.4}$ | 68.8 $_{\pm0.9}$ | 68.6 $_{\pm1.5}$ | 62.58 | 67.64 |

## G  Potential ethical and social implications

There are a few potential ethical considerations that could arise from the molecular graph representation learning methods discussed in :

**Bias and fairness**: We recognise that the molecular datasets used for pre-training and benchmarking may inadvertently encode biases or lack diversity, which could lead to models that unfairly under-

Table 18: **Benchmarking topological properties, on 2M ZINC15.** We report the mean square error or the cross entropy on eight datasets (*i.e.*, smaller is better).

| | Node | | | Pair | | | Graph | | | |
|---|---|---|---|---|---|---|---|---|---|---|
| | Degree | Cent. | Cluster | Link | Jaccord | Katz | Diameter | Conn. | Cycle | Assort. |
| RANDOM | 0.001 | 0.008 | 0.003 | 0.078 | 0.012 | 0.017 | 177.924 | 0.087 | 2.933 | 0.029 |
| ATTRMASK | 0.058 | 0.010 | 0.003 | 0.082 | 0.013 | 0.020 | 142.832 | 0.071 | 3.232 | 0.027 |
| GRAPHCL | 0.063 | 0.009 | 0.003 | 0.079 | 0.018 | 0.022 | 87.426 | 0.064 | 1.716 | 0.017 |
| GRAPHMAE | 0.032 | 0.011 | 0.004 | 6.885 | 0.215 | 0.038 | 123.677 | 0.066 | 3.094 | 0.025 |
| RGCL | 0.356 | 0.011 | 0.003 | 0.075 | 0.013 | 0.017 | 124.869 | 0.067 | 3.528 | 0.029 |

perform on certain data subsets. We have made efforts to use representative datasets, but further testing on diverse molecules and monitoring for fairness issues remain important future directions.

Relatedly, an inherent risk with data-driven methods is "stereotyping" along chemical lines, where models reinforce historical assumptions. We believe conscious testing on new classes of molecules can mitigate this. The goal is to build broad molecular understanding.

**Dual use**: As with many advances in chemistry ML, dual use concerns exist. While we aim to accelerate beneficial therapeutics, we strongly discourage misuse of these methods to design harmful substances. We support efforts toward responsible AI practices.

**Environmental impact**: Training large molecular graph models consumes significant computational resources and energy, efforts should be made to minimise the environmental impact, *e.g.*, by using efficient methods and carbon-neutral compute.

**Privacy**: While molecular graph data is less sensitive than data about individuals, any efforts to link models back to original private datasets should be handled carefully.

Overall, we make efforts to align research practices with principles of sharing benefits, avoiding harm, mitigating biases, and practising transparency. We welcome feedback from the community as we continue working to address these ethical dimensions. Our goal is innovation that aligns with broad social good.

