# OpenReview forum: "Evaluating Self-Supervised Learning for Molecular Graph Embeddings"
_NeurIPS.cc/2023/Track/Datasets_and_Benchmarks — NeurIPS 2023 Datasets and Benchmarks Poster_

### Official Review · Reviewer_cQn4 · 2023-07-14

**Rating:** 7
**Confidence:** 4
**Correctness:** Yes
**Clarity:** Yes

**Strengths:**

- Graph Self-Supervised Learning is an important problem.

- This paper is well organized and clearly written. The technical details are also easy to follow.

- The experiment are extensive and the results are interesting.

**Additional Feedback:**

N/A

**Documentation:**

Yes

**Opportunities For Improvement:**

- More GCL baselines should be included or discussed in this paper, e.g., [1,2,3].

- More detailed examples in Molecular Graphs should be presented instead of CV examples, e.g., Figure 3.

- I suggest authors can release their codes to public.

- How about the efficiency of these methods?


[1] CLEAR: Cluster-Enhanced Contrast for Self-Supervised Graph Representation Learning, TNNLS 22

[2] Prototypical Graph Contrastive Learning, TNNLS 22

[3] Augmentation-Free Graph Contrastive Learning of Invariant-Discriminative Representations, TNNLS 23

**Relation To Prior Work:**

Yes

**Summary And Contributions:**

This paper evaluates graph Self-Supervised Learning (GSSL) for molecular graph embeddings. The code is provided.

---

> ### Author Response · Authors · 2023-08-17
> **Response**
>
> Thank you for your insightful feedback.
> - We've incorporated references to the suggested GCL baselines and GSSL methods in Appendix B.2.
> - We concur with your suggestion and have substituted the CV examples with abstract shapes in Figure 3.
> - Please note that the code was made publicly accessible at the time of our initial submission.
> - Considering the uniformity of both the finetuning and probing processes – both utilise the same GNN backbone.
> - We have documented and compared the runtime and number of parameters for each GSSL method during the pre-training phase in Appendix B.5.

---

> > ### Comment · Reviewer_cQn4 · 2023-08-29
> >
> > Thanks for the response. I cannot see the discussion of references in Appendix B.2. Is there any version problem?

---

> > > ### Author Response · Authors · 2023-08-29
> > > **Revised appendix is also in the main pdf**
> > >
> > > Dear Reviewer cQn4,
> > >
> > > We have included the revised appendix within the main PDF file rather than in the Supplementary Material section.

---

> > > > ### Comment · Reviewer_cQn4 · 2023-08-29
> > > >
> > > > Thanks. Can you please point this out? Which line?

---

> > > > > ### Author Response · Authors · 2023-08-29
> > > > >
> > > > > Discussions on the graph masked autoencoders are in lines 738-740, and we provide general discussions in the entire Appendix B.2 (lines 724-755).

---

> > > > > > ### Comment · Reviewer_cQn4 · 2023-08-29
> > > > > >
> > > > > > It seems that you miss my referred GCL baselines for detailed comparison and discussion. Actually I didn't mention graph masked autoencoders. Please read my review.

---

> > > > > > > ### Author Response · Authors · 2023-08-29
> > > > > > >
> > > > > > > Thank you for your kind reminder. The current revised version includes a comprehensive discussion of these techniques in see Lines 743-748. Please let us know if we miss anything.

---

> > > > > > > > ### Comment · Reviewer_cQn4 · 2023-08-29
> > > > > > > >
> > > > > > > > Thanks for your updating. I will raise my rating to 7.

---

### Official Review · Reviewer_rFyK · 2023-07-20

**Rating:** 6
**Confidence:** 4
**Correctness:** To the best of my understanding, the …
**Clarity:** The paper is well-written.

**Strengths:**

1. The paper is generally well-structured and easy to understand.
2. The authors have effectively summarized and presented their primary findings.
3. The authors back their findings with a variety of experiments, providing both qualitative and quantitative support.

**Additional Feedback:**

This paper's experiments and findings are valuable contributions to the community. I have a few minor comments:

1. I recommend swapping the order of the two paragraphs found in lines 216-221 and 222-227.
2. A more detailed explanation is needed for Table 3 to support the claim made in lines 216-221.
3. Table 2 should be placed closer to line 255, where it is first referenced. The same adjustment should be made for Table 4.

**Documentation:**

The authors have provided detailed information about their experiments.

**Ethics:**

No ethical concerns have been identified.

**Limitations:**

The authors effectively delineate the scope of this research.

**Opportunities For Improvement:**

1. Even with limited space, it would be beneficial to include at least brief descriptions of the GSSL methods employed (either within the paper or in an appendix), as these are critical for comprehending the paper's primary findings.
2. As a comprehensive benchmark paper, it would be useful to see the authors derive new insights and suggest future research directions for GSSL, based on their findings. Such insights add considerable value to meta-analysis studies like this one.
3. Findings 2 and 3 appear somewhat trivial in comparison to the others. Few researchers would expect a perfect match between rankings (and optimal hyperparameters) in probing and fine-tuning tasks. A deeper discussion on these findings would be valuable, even if it is designated for future work, as suggested by the authors in lines 134-135.

**Relation To Prior Work:**

As a benchmark paper comparing methods from numerous previous works, a more thorough discussion on how this paper's findings relate to partial observations and findings in previous research would be valuable.

**Summary And Contributions:**

Graph Self-Supervised Learning (GSSL) allows for the extraction of node/graph embeddings from molecular graphs without requiring labels. However, due to the broad applicability of most GSSL methods across numerous downstream tasks, evaluating their performance can be challenging. This paper introduces "Molecular Graph Representation Evaluation" (MolGraphEval), which includes a suite of probing tasks designed to tackle this issue. The authors employ MolGraphEval to benchmark current GSSL methods and discover that existing evaluation methodologies fail to adequately encompass the full complexity of the task landscape.

---

> ### Author Response · Authors · 2023-08-17
> **Response**
>
> [**Brief description of employed GSSL methods**]
>
> Thanks for the comments! We've included more descriptions of these GSSL methods in Appendix B.2.
>
> [**New insights and suggestions for future works**]
>
> Thank you for raising this. We have added a few more discussions on practical guides for future works in Appendix B.6.
>
> [**Deeper discussion on Findings 2 and 3**]
>
> Thank you for this suggestion. We agree that diving deeper on the rankings / hyperopts between probing and fine-tuning would be useful, we have added a few more discussions in Appendix B.6.
>
> [**Relation to prior work**]
>
> Thank you for this suggestion. We have added discussions on related works in Appendix B.2.
>
> [**Additional feedbacks**]
>
> Thank you for these great suggestions. We’ve swapped the two paragraphs to make more coherent flows; adding more interpretations on Table 3; and adjusted the locations of Tables 2 & 4.

---

> > ### Comment · Reviewer_rFyK · 2023-08-30
> > **Comment**
> >
> > Thank you for your response. I appreciate the additions you made.

---

### Official Review · Reviewer_xXbt · 2023-07-21
**Reviews to paper #456**

**Rating:** 7
**Confidence:** 4

**Strengths:**

* The proposed MOLGRAPHEVAL method offers a more nuanced and comprehensive evaluation of GSSL methods for molecular graph embeddings, which can help researchers better understand the strengths and weaknesses of these methods. This can lead to the development of more effective and efficient methods for molecular property prediction, which has important applications in drug discovery and other areas.

* The paper addresses an important problem in the field of molecular property prediction, which has broad applications in drug discovery and other areas. The proposed method can be applied to a wide range of GSSL methods and can help researchers better understand the performance of these methods.

Ethical and social implications are not relevant in the work.

**Additional Feedback:**

N/A

**Clarity:**

The paper is well written and easy to understand. The authors have provided a detailed description of their methods, which makes it easy to reproduce their results.


**Correctness:**

The paper is a benchmark submission, and the evaluation methods and experiment design are appropriate and performed correctly. The authors have also provided a detailed description of their methods, which makes it easy to understand and reproduce their results.
However, there still a couple of minor issues that could be addressed:

* One of the MOLGRAPHEVAL is the generic graph properties, however, it is unclear to me that why such topological properties are associated with the benchmark methods? Such graph peoperties are independent to algorithms of GSSL, unless the methods themselves are predicting the properties. I don't find the explanation in Table 2 and also in the Section 5.1.
* The definition of node centrality is inconsistent with its mathematical formulation. The equation is essentially the "eigenvector centrality" [1], which is a measure of the influence of a node in a network. However, the authors define it as the "average centrality of all nodes in the graph", which is not the case. The authors should either change the definition or the equation. Also, include the definition of $\lambda$ if it is revised accordingly.
* In the section of generic graph properties, the "Node" and "Pair" are defined on individual node or edge. What are reported in Table 2, average or sum of all nodes or edges? It is not clear to me. Please explain.

[1] Zaki, Mohammed J.; Meira, Jr., Wagner (2014). Data Mining and Analysis: Fundamental Concepts and Algorithms. Cambridge University.

**Documentation:**

N/A

**Limitations:**

The authors briefly mention the importance of ethical and responsible use of these methods, but they do not provide a detailed discussion of the potential negative societal impact of their work. While the paper focuses primarily on technical aspects of molecular property prediction, it is important to consider the broader implications of this research for society as a whole.

To address this limitation, the authors could include a more detailed discussion of the potential ethical and social implications of their work, including issues related to bias, fairness, and privacy. They could also consider conducting a more in-depth analysis of the potential negative societal impact of their work, and provide recommendations for how to mitigate these risks.


**Opportunities For Improvement:**

* Limited scope of evaluation: The authors only evaluate a limited number of GSSL methods and downstream tasks, which may not fully capture the performance of these methods in all scenarios. Future work could expand the scope of evaluation to include more methods and tasks.

* Lack of comparison with other evaluation methods: The authors do not compare their proposed method with other existing evaluation methods, which makes it difficult to assess the relative strengths and weaknesses of different methods.

* Limited discussion of ethical and social implications: While the paper briefly mentions the importance of ethical and responsible use of these methods, it does not provide a detailed discussion of the potential ethical and social implications of these methods. Future work could explore these issues in more detail.

**Relation To Prior Work:**

As a benchmark paper, it will be better to provide the descriptions of algorithms used in the MolGraphEval, with a categorized properties of those algorithms would be a plus. For example, for example, the base model, number of parameters (model size), primary training techniques, etc.


**Summary And Contributions:**

The paper proposes a method called "Molecular Graph Representation Evaluation" (MOLGRAPHEVAL) to evaluate the performance of Graph Self-Supervised Learning (GSSL) methods for molecular graph embeddings. The authors argue that current evaluation methodologies fail to capture the entirety of the landscape, and MOLGRAPHEVAL offers a suite of probing tasks grouped into three categories: (i) generic graph, (ii) molecular substructure, and (iii) embedding space properties. By benchmarking existing GSSL methods against both current downstream datasets and their suite of tasks, the authors uncover significant inconsistencies between inferences drawn solely from existing datasets and those derived from more nuanced probing.

---

> ### Author Response · Authors · 2023-08-17
> **Response**
>
> [**Limited scope of evaluation**]
>
> Thank you for your suggestion. In this paper, we focus exclusively on molecules, which allows us to consider molecule-specific attributes, such as molecular substructures in Section 5.2. We agree that an analysis of GSSL methods for other modalities would be very interesting, and we leave this to future work.
>
> [**Compare with other evaluation frameworks**]
>
> Thank you for this suggestion. We discussed relevant works that use probe models to analyse self-supervised learning on other modalities in Section 2. To the best of our knowledge, we are the first to use probe models to understand graph self-supervised learning, and there are follow-ups [1, 2, 3] that acknowledge our contribution. In the revised manuscript, we have discussed other works that analyse graph self-supervised learning in Appendix B.2. Please let us know if anything remains missing.
>
> [**Discussion on the ethical and social implications**]
>
> Thank you, we have included a section as Appendix G on the ethical considerations from different lenses including: 1) bias and fairness; 2) dual use; 3) environment impact, and 4) privacy.
>
> [**Why evaluating topological properties**]
>
> Thank you for raising this. Certain GSSL methods, like Cont.Pred [ref. 10], are indeed fashioned around general topological metrics to predict node properties based on its K-hop neighbourhood subgraph. Given that these generic graph properties have been employed in chemical informatics for decades [refs. 52-55] for molecular property prediction, our primary aim is to examine the efficacy of the knowledge acquired post each GSSL method.
>
> [**Definition of node centrality**]
>
> Thank you for raising this. We essentially use the same definition of node’s eigenvector centrality from [4] (Chapter 2.1.1), as well as the code implementation:
>
> *centrality = nx.eigenvector_centrality(graph_nx, max_iter=1500)*
>
> Thank you for guiding us to other variations of centrality (eccentricity, closeness, betweenness) from [5] (Chapter 1.4.3). In the original manuscript, we wrote as “proportional to the average centrality of its neighbours”, where the proportion constant \lambda is essentially an eigenvector of the adjacency matrix A.
>
> [**Calculation details of node and pair properties**]
>
> Thank you for raising this. All the node properties are calculated based on each node, and we train the probe model to predict each node metric using pre-trained embeddings. As for the pair properties, we randomly sample a fixed number (i.e., 10) of atomic pairs for a given molecular graph, then split them and train the probe models, all pairs from one molecular graph are ensured to be in the same split. We have clarified this implementation detail in the revised manuscript.
>
> [**Descriptions of the algorithms and the base model**]
>
> Agreed, thank you for this suggestion! We have added this to Appendix B.2 in the revised manuscript.
>
> **Reference**
>
> [1] Probing Graph Representations, AISTATS 2023
>
> [2] Enhancing Activity Prediction Models in Drug Discovery with the Ability to Understand Human Language, arXiv 2023
>
> [3] Exposing the Limitations of Molecular Machine Learning with Activity Cliffs, JCIM 2022
>
> [4] Graph Representation Learning, William Hamilton, (2020)
>
> [5] Data Mining and Analysis: Fundamental Concepts and Algorithms. Zaki, Mohammed J.; Meira, Jr., Wagner (2020 2nd Edition). Cambridge University.

---

> > ### Comment · Reviewer_xXbt · 2023-08-26
> > **Response to Authors**
> >
> > Thanks for providing the thoughtful responses to my comments.
> > After careful consideration, I am pleased to conclude that your paper is well-written, well-structured, and makes a significant contribution to the field of graph self-supervised learning, especially in bioinformatics and drug discovery working on molecules.
> >
> > The new appendices have addressed my concerns on ethical reviews, which is an essential aspect of responsible research. Your discussion of the ethical implications of GSSL methods is comprehensive and thought-provoking, and it highlights the importance of considering the potential consequences of such methods in practice.
> >
> > The revised version also clarified the rationale behind evaluating topological properties and the definition of node centrality. I appreciate the references you have provided to support your arguments and the details you have shared about the calculation of node and pair properties.
> >
> >
> > To this end, I will keep my original score, which is a relatively high score now, I wish you continued success in your endeavors.

---

> > > ### Author Response · Authors · 2023-08-26
> > > **Thank you!**
> > >
> > > Dear Reveiwer xXbt,
> > >
> > > We are pleased to hear that the revised version of the paper and the new appendices have effectively addressed your concerns. And we will keep work in these fields and share our findings with the community.

---

### Official Review · Reviewer_qGTN · 2023-07-21
**Evaluating Self-Supervised Learning for Molecular Graph Embeddings**

**Rating:** 6
**Confidence:** 5

**Strengths:**

1.	This work introduces "Molecular Graph Representation Evaluation (MolGraphEval)," which presents a suite of probing tasks classified into three categories: generic graph properties, molecular substructure properties, and embedding space properties.
2.	The study conducts extensive experiments involving 90,918 probe models and 1,875 pre-trained GNNs (for nine GSSL models) to provide an unbiased evaluation of molecular graph embeddings obtained through GSSL methods.
3.	The research identifies several intriguing findings that could assist the community in designing improved GSSL algorithms.

**Additional Feedback:**

Please see the Opportunities for Improvement section for details.

**Clarity:**

The structure and writing are good. However, the authors are encouraged to make the information self-contained by providing definition or detailed explanations for Prob model and MPP abbreviation.

**Correctness:**

Partially, from an evaluation methods perspective, several state-of-the-art generative baselines are absent. Additionally, the experiment design raises concerns as the supervised probing task may not be adequately aligned with the purpose of self-supervised learning. Furthermore, the use of limited datasets in this paper warrants attention.

**Documentation:**

Yes.

**Ethics:**

No.

**Opportunities For Improvement:**

1.	While this work aims to benchmark GSSL, the experiments adopt a "supervised" setting (80%/10%/10% splits for the training/validation/testing set) to test performance on downstream tasks. More experiments under semi-supervised or even few-shot settings would be valuable to build a comprehensive understanding.
2.	The definition of the Probe model used in this paper should be clearly explained to make the content self-contained. The difference between the Probe model and the traditional linear probe based on simple MLPs is unclear. In section 3, it seems that MLPs with one hidden layer (also known as a linear probe in the graph domain) are also referred to as Probe models. More explanations regarding the definition of the Probe model are required.
3.	In table 1, the comparison is insufficient. In addition to the intuitive Random baseline, a supervised GNN model serves as another strong competitor. Including the supervised GNN baseline would help us understand if graph self-supervised learning makes real progress under the current supervised scenarios.
4.	Different from the existing benchmark [16], which provides a benchmark for self-supervised learning on graphs, this paper additionally considers generative models for evaluation. However, the generative baselines included are rather limited. For example, state-of-the-art graph generative models (e.g., GraphMAE [1], MaskGAE[2], and MGAE[3]) are missing.
5.	To increase readability, it would be better to group the baseline methods in Table 1 into different groups, such as contrastive and generation approaches. Additionally, since we have two types of self-supervised learning baselines, analyzing their differences in the new evaluation scenario would be interesting.
6.	The idea to evaluate GSSL methods from the graph property perspectives, in addition to the MPP tasks, is commendable. However, the paper lacks insights on how/when to leverage these graph property tasks to benefit existing GSSL methods.
7.	While the authors claim that they follow previous work [10, 11] to choose datasets for benchmarking, many datasets used in [11] (e.g., PROTEINS, DD, and COLLAB) are not considered in this work.

[1] GraphMAE: Self-Supervised Masked Graph Autoencoders, KDD, 2022.

[2] MaskGAE: Masked graph modeling meets graph autoencoders, arxiv, 2022.

[3] MGAE: Masked Autoencoders for Self-Supervised Learning on Graphs, arxiv, 2022.

**Relation To Prior Work:**

Yes.

**Summary And Contributions:**

Graph self-supervised learning (GSSL) has demonstrated significant success in acquiring informative graph representations without relying on labeled samples. However, existing benchmarks predominantly concentrate on two aspects: 1) evaluating the performance of current GSSL methods using the MPP tasks, and 2) contrastive-based GSSL approaches. To overcome these limitations, this work introduces a comprehensive set of evaluation tasks that encompass various levels of graph properties. It benchmarks both contrastive- and generation-based GSSL algorithms against these tasks. Empirical experiments conducted with the new evaluation metrics unveil several intriguing findings.

---

> ### Author Response · Authors · 2023-08-17
> **Thank you for your feedback! Response - Part 1**
>
> [**Benchmark with semi-supervised or few-shot settings**]
>
> Thank you for raising this. We believe there is a slight misunderstanding. The approach of using unsupervised or self-supervised pre-training on large unlabeled datasets, followed by supervised fine-tuning or probing on downstream datasets, is a well-established paradigm in vision [1,2,3], language [4,5,6], and graph studies [7, 8]. We'll delve deeper into the probing settings in our subsequent point. In our revised codebase, we've added features pretaining to few-shot learning and also included references to leading semi-supervised learning methods in the documentation. This update offers future practitioners a more robust toolkit to explore.
>
> [**Definition of probe models**]
>
> Thank you for the suggestion. Recall the refs.16-31 in Section 2, the most common definition of a probe model can be combined based on the description from (Alain and Bengio, Ref. 16) and (Hewitt & Manning, Ref. 18):
>
> *A probe is a linear classifier designed to assess the quality of information encapsulated within a linear transformation of a neural network's representational space. It aids researchers in understanding the functions and dynamics of intermediate layers and pre-training methodologies, offering insights into model behavior and highlighting potential issues.*
>
> Therefore, in most research, a supervised setting is employed to learn this linear transformation, with some studies exploring a zero-shot setting (Hendricks et al., Ref. 22). The architectures of probe models, however, lack uniformity across previous literature. Many prior studies have opted for linear models [refs. 16-18, 20-31], which employ a single set of linear weights and additive biases from embeddings to metrics without hidden layers. Occasionally, MLPs [refs. 19, 31] or more intricate designs [ref. 22] are used.
>
> Regarding the detailed set-ups in our paper, we describe these in Section.3 (lines 96-99):
>
> *We mainly compute and compare the quality of pre-trained embeddings using linear probe models. We have also experimented MLPs with one hidden layer as the probe models, as this architecture is utilised in some previous works. We observe similar findings with both probe architectures and reported the results of MLP probes in Appendix B.4.*
>
> Please let us know if anything remains unclear.
>
> [**Adding results for models that are supervisedly trained on downstream data**]
>
> Thank you for your suggestion. We considered this setting at the beginning of our study and conducted some experiments. However, we decided not to use these results for the following reasons:
> 1. The primary focus of this study is to evaluate and benchmark existing GSSL methods using designed metrics.
> 2. The number of combinations of (supervised pre-training dataset(s), downstream dataset(s)) is at least $N^2$, which is computationally expensive to explore, yet the additional learned knowledge is very limited;
> 3. Current frameworks are closer to the real-world setting, where we often have access to a single large data collection for pre-training and multiple downstream datasets.
>
> Our current framework already allows such directions, i.e., change the pre-training dataset as one of the downstream dataset. We have added clarification on this point in Appendix F, and we will also note this in the documentation to facilitate practitioners to conduct such explorations.
>
> [**Implementing more baselines**]
>
> Thanks for the reminder. We have indeed included the implementations of GraphMAE in the code base and also conducted some experiments with the reported optimal hyperparameters in Appendix F. We have now cited and acknowledged the contributions of MaskGAE and MAGE in the direction of masked graph autoencoders in Appendix B.2.
>
> [**Comparison within contrastive and generative groups**]
>
> Thank you for this suggestion. We have included discussions on the general comparison between the contrastive and generative GSSL methods in Appendix B.2.
>
> [**How/when to leverage graph property tasks to benefit existing GSSL methods**]
>
> We believe that our results can benefit existing GSSL methods in several ways. For example, (i) because randomised features outperform almost every GSSL method on node- and pair-level metrics, augmenting GSSL-learned features with randomised ones can improve performance; (ii)  . Sorry if this was not clearer, we made it more explicit in the revised manuscript.
>
> [**Use more datasets for pre-training**]
>
> Thank you for that suggestion! We would like to point out that the PROTEINS, DD, and COLLAB datasets do not contain molecular graphs. In this work, we focus on molecules, which allows us to consider molecule-specific attributes, e.g., molecular substructures in Section 5.2. We agree that analysing GSSL methods for other modalities would be very interesting, and we leave this to future work.
>
> [**MPP abbreviation**]
>
> The abbreviation is now added in line 21 of the revised manuscript; thank you for pointing this out!

---

> ### Author Response · Authors · 2023-08-17
> **Response - Part 2**
>
> **Reference**
>
> [1] A Simple Framework for Contrastive Learning of Visual Representations, Ting Chen et al., ICML 2020
>
> [2] Momentum Contrast for Unsupervised Visual Representation Learning, Kaiming He et al., CVPR 2020
>
> [3] Masked Autoencoders Are Scalable Vision Learners, Kaiming He et al., CVPR 2022
>
> [4] BERT: Pre-training of Deep Bidirectional Transformers for Language Understanding, Jacob Devlin et al., NAACL 2019
>
> [5] Language Models are Few-Shot Learners, Tom Brown et al., NeurIPS 2020
>
> [6] Biological structure and function emerge from scaling unsupervised learning to 250 million protein sequences, Alex Rives et al., PNAS 2021
>
> [7] Strategies for Pre-training Graph Neural Networks, Weihua Hu et al., ICLR 2020
>
> [8] Pre-training Molecular Graph Representation with 3D Geometry, Shengchao Liu et al., ICLR 2022

---

> ### Author Response · Authors · 2023-08-25
> **Inquiry for Feedback**
>
> Dear Reviewer qGTN,
>
> I hope this message finds you well. We would like to begin by expressing our sincere gratitude for your invaluable feedback on our submission. Rest assured, we have carefully considered your insights and have incorporated them into our revised manuscript.
>
> We are committed to addressing all of your concerns comprehensively. If there are additional questions or comments you would like us to consider, we welcome your input. Your expertise is greatly appreciated, and we aim to ensure that all aspects of our work meet the highest standards of clarity and rigor.
>
> We are deeply grateful for the time and effort you have invested in the review process. Any further feedback from you will certainly contribute to enhancing the quality of our work.
>
> Thank you once again for your continued involvement and guidance. We eagerly await your additional feedback.
>
> Warm regards,
> Authors

---

> > ### Comment · Reviewer_qGTN · 2023-08-29
> >
> > Thank you for your detailed reply. The author's rebuttal addressed my concerns, so I would like to improve my score.

---

> > > ### Author Response · Authors · 2023-08-29
> > > **Thank you!**
> > >
> > > Dear qGTN,
> > >
> > > We greatly appreciate your acknowledgment of the enhancements made, as evidenced by the increased scores!

---

### Official Review · Reviewer_Naos · 2023-07-22

**Rating:** 6
**Confidence:** 4

**Strengths:**

 - This work conducts an extensive evaluation of nine graph self-supervised learning methods. Besides downstream molecular property prediction tasks, this work collected 18 properties as probing tasks and explored different aspects of GSSL methods.
- On generic graph property prediction, this work revealed that random initialized models are better at predicting node-level and edge-level tasks.
- On embedding space property evaluation, this work discovered that uniformity is not always a strong indicator for molecular property prediction performance. Moreover, this work provides evidence that GSSL methods generate better embedding in terms of these properties.

**Additional Feedback:**



**Clarity:**

 - For the results of benchmarking topological property predictions (Table 2) and substructure prediction (Table 5), it would be better to provide the average ground-truth values of the properties, such as average degree, which makes it more clear to interpret the improvement of prediction results.
- How does this work evaluate the uniformity of the pre-trained embeddings?
- For the alignment evaluation in Figure 6, what do the x-axis and y-axis indicate? How should one interpret the difference between the results of AttrMask and GraphCL?

**Correctness:**

 To the review's best knowledge, the evaluation methods and experiments are performed correctly.

**Documentation:**

This work provides its implementation and documents for hyper-parameter settings. It would be better to also provide the documents of reproducing the main experiments and accessing the probing task data, such as the substructure property prediction.

**Ethics:**

I have no ethical concerns with this paper.

**Limitations:**

 This work has discussed the limitations.

**Opportunities For Improvement:**

 - It would be better to provide a more detailed discussion of the self-supervised learning methods. For example, the taxonomy of the nine GSSL methods. Based on the taxonomy, it would be better to discuss whether there is a trend of a certain type of GSSL method being good at a certain task.
- In the evaluation of generic graph property prediction, it would be better to explore whether a combination of two types of GSSL methods can lead to better node-level or edge-level property prediction.

**Relation To Prior Work:**

 The relation to prior work is clearly discussed.

**Summary And Contributions:**

This work conducts an extensive evaluation of nine graph self-supervised learning (GSSL) methods for downstream molecular property prediction tasks in terms of linear probing and fine-tuning. The GSSL methods include EdgePred, AttrMask, GPT-GNN, InfoGraph, ContextPred, GROVER, GraphCL, JOAO, and GraphMVP. The downstream tasks are eight molecular property prediction tasks from MoleculeNet. Furthermore, this work probes the performance of using the pre-trained embeddings to predict: (i) generic graph properties, such as node degree, (ii) molecular substructure, and (iii) embedding space property, such as uniformity. Overall, this work designed 18 probing tasks to assess the quality of GSSL methods.

---

> ### Author Response · Authors · 2023-08-17
> **Thank you for your feedback! Our Response**
>
> [**More detailed discussions on the compared methods**]
>
> Thank you. We have included a detailed discussion of the benchmarked GSSL methods in Appendix B.2.
>
> [**Combining multiple pre-training objectives**]
>
> Thank you for your valuable suggestion. We decided to not include them in the current paper because they would massively increase the number of methods to consider and experiments to run:
> - It would lead to an exponential increase in the hyperparameter search space compared to our current explorations.
> - It would require adding many more methods to ensure a fair comparison among them. Beyond straightforward methods such as the linear combination of weighted losses, the domain of multi-task learning boasts a plethora of well-established techniques for weight balancing. This encompasses solutions for conflicting gradients [1, 2], disparities in gradient magnitudes [3, 4, 5], and imbalanced loss scales [6, 7, 8].
>
> However, we are committed to  provide our code repository, which allows one to easily integrate multiple pre-training tasks using preset fixed weights. Thereby, we aim to  serve a good starting point for future research on such.
>
> [**Ground true values on topological metrics and substructures**]
>
> Thank you for your feedback! To enhance clarity, we've incorporated the average ground true values alongside each bar plot pertaining to topological metrics in Appendices D.4 and E.1. For a comprehensive understanding, interpretations of these additions are provided in the relevant sections.
>
> [**Evaluation of the uniformity**]
>
> We adopted the procedures outlined in [9], and more specifically, relied on the code snippet from [10] to assess the uniformity of the pre-trained embedding space. A higher uniformity value suggests that the embedding vectors are distributed more uniformly. Conversely, a lower value indicates less uniform distribution.
>
> [Interpretation on Figure 6]
> Thank you for raising this. On the graph, the x-axis illustrates the cosine similarity between two molecular embeddings ($X_a, X_b$), and the y-axis denotes their respective counts. The cosine similarity is defined as:
>
> $S_C(A, B) = \cos (\theta) = \frac{\mathbf{X_a} \cdot \mathbf{X_b}}{|\mathbf{X_a}||\mathbf{X_b}|} $
>
> This similarity metric spans a range from $-1$ (indicating exact oppositeness) to $1$ (signifying identicalness). A value of $0$ points to orthogonality or decorrelation, and intermediate values capture varying degrees of similarity or dissimilarity. Within a specific dataset, if all properties of a molecular pair are identical, they are categorised as positive pairs (denoted as ‘pos’ in the legend). Otherwise, they are labelled negative (represented as ‘neg’ in the legend).
>
> The core objective of this panel is to probe how similarity metrics fluctuate across different GSSL methodologies. When contrasting AttrMask and GraphCL on the BBBP and Tox21 datasets, it's evident that AttrMask embeddings yield higher similarity measures for both negative and positive pairs. However, this doesn't implicitly suggest superiority over GraphCL. Importantly, both these methods exhibit a sharper distinction between positive and negative pairs compared to randomised embeddings.
>
> [**Documentation on reproducing the results**]
>
> Thank you for highlighting this aspect. In our initial submission, we already offered the code, pre-trained weights, bash scripts, and pertinent documentation to facilitate reproducing our paper's results.
>
> **Reference**
>
> [1] Zhao Chen et al., Just pick a sign: Optimising deep multitask models with gradient sign dropout, NeurIPS 2020
>
> [2] Tianhe Yu et al., Gradient surgery for multi-task learning, NeurIPS 2020
>
> [3] Liyang Liu et al., Towards impartial multi-task learning, ICLR 2021
>
> [4] Aviv Navon et al., Multi-task learning as a bargaining game, ICML 2022
>
> [5] Ozan Sener and Vladlen Koltun. Multi-task learning as multi-objective optimization, NeurIPS 2018
>
> [6] Zhao Chen et al., Gradnorm: Gradient normalisation for adaptive loss balancing in deep multitask networks, ICML 2018
>
> [7] Michelle Guo et al., Dynamic task prioritisation for multitask learning, ECCV 2018
>
> [8] Alex Kendall et al., Multi-task learning using uncertainty to weigh losses for scene geometry and semantics, CVPR 2018
>
> [9] Tongzhou Wang and Phillip Isola, Understanding contrastive representation learning through alignment and uniformity on the hypersphere, ICML 2020
>
> [10] https://github.com/ssnl/align_uniform/blob/master/align_uniform/__init__.py#L4-L9

---

### Author Response · Authors · 2023-08-17
**General comment during rebuttal**

We extend our sincere gratitude to all reviewers and ACs for their invaluable insights and assistance in enhancing this paper. The manuscript has been revised in accordance with their feedback, with all modifications highlighted in red. Below, we provide detailed responses to each of the comments made by the reviewers.

---

### Author Response · Authors · 2023-08-21
**Request for re-evaluations**

Dear reviewers,

Thank you for your thoughtful review of our manuscript. We've taken your feedback and have made revisions accordingly. We kindly request that you review our responses, as we believe they directly address the concerns you raised. If you find our revisions and clarifications satisfactory, we humbly ask that you consider reflecting these changes in your evaluation scores, if appropriate. Your expertise and continued input are highly valued, and we look forward to hearing your thoughts.

---

### Decision · Program_Chairs · 2023-09-22

**Decision:**

Accept (Poster)

**Comment:**

This paper extensively evaluates nine graph self-supervised learning (GSSL) methods for molecular property prediction tasks, including probing generic graph properties, molecular substructure, and embedding space property using 18 probing tasks. It introduces "Molecular Graph Representation Evaluation" (MOLGRAPHEVAL) to address limitations in existing GSSL evaluation methodologies, highlighting inconsistencies in current evaluations and offering a more nuanced assessment of GSSL methods, especially in molecular graph embeddings. The reviewers have raised several significant concerns but the author's rebuttal has provided thorough and in-depth responses to most of the concerns. The paper is a borderline one based on the final scores (compared to other submissions) but I am inclined towards acceptance.